# Impact of environmental changes and land-management practices on wheat production in India

**Shilpa Gahlot[1], Tzu-Shun Lin[2], Atul K Jain[2], Somnath Baidya Roy[1], Vinay K Sehgal[3], Rajkumar Dhakar[3]**

[1]Centre for Atmospheric Science, Indian Institute of Technology Delhi, New Delhi, 110016 India,

[2]Department of Atmospheric Science, University of Illinois, Urbana, IL, 61801 USA,

[3]Department of Agricultural Physics, Indian Agricultural Research Institute, New Delhi, 110012, India

*Correspondence to*: Somnath Baidya Roy (drsbr@iitd.ac.in)

**Abstract.** Spring wheat is a major food crop that is a staple for a large number of people in India and the world. To address the issue of food security, it is essential to understand how productivity of spring wheat changes with changes in environmental conditions and agricultural management practices. The goal of this study is to quantify the role of different environmental factors and management practices on wheat production in India in recent years (1980 to 2016). Elevated atmospheric $CO_2$ concentration ($[CO_2]$) and climate change are identified as two major factors that represent changes in the environment. The addition of nitrogen fertilizers and irrigation practices are the two land-management factors considered in this study. To study the effects of these factors on wheat growth and production, we developed crop growth processes for spring wheat in India and implemented them in the Integrated Science Assessment Model (ISAM), a state-of-the-art land model. The model is able to capture site-level observed crop leaf area index (LAI) and country scale production. Numerical experiments are conducted with the model to quantify the effect of each factor on wheat production on a country scale for India. Our results show that elevated $[CO_2]$ levels, water availability through irrigation and nitrogen fertilizers have led to an increase in annual wheat production at 0.67, 0.25 and 0.26 Mt yr$^{-1}$, respectively, averaged over the time period 1980-

2016. However, elevated temperatures have reduced the total wheat production at a rate of 0.39 Mt yr$^{-1}$ during the study period. Overall, the $[CO_2]$, irrigation, fertilizers, and temperature forcings have led to 22Mt (30%), 8.47 Mt (12%), 10.63 Mt (15%) and -13 Mt (-18%) changes in countrywide production, respectively. The magnitudes of these factors spatially vary across the country thereby affecting production at regional scales. Results show that favourable growing season temperatures, moderate to high fertilizer application, high availability of irrigation facilities, and moderate water demand make the Indo-Gangetic plain the most productive region while the arid northwest region is the least productive due to high temperatures and lack of irrigation facilities to meet the high water demand.

**1 Introduction**

Wheat is a major food crop, ranked third in India and fourth in the world in terms of its production (FAOSTAT, 2019). Wheat can be of two main types: winter and spring wheat. Winter wheat undergoes a 30-40 day long vernalization period induced by below-freezing temperatures and hence has a longer growing season of 180-250 days. In contrast, spring wheat, which does not undergo vernalization, has a growing season of 100-130 days (FAO Crop Information, 2018). In India, spring wheat is sown during October-November and harvested during February-April (Sacks et al., 2010). It is grown in widely divergent climatic conditions across the country where different environmental factors like temperature, water availability, and $[CO_2]$ may affect growth and yield. Ideally, a daily average temperature range of 20-25 °C is ideal for wheat growth (MOA, 2016). Studies have reported heat stress in wheat for temperatures between 25 °C to 35 °C (Deryng et al., 2014) during the grain development stages. Beyond the temperatures of 35 °C, wheat fails to survive. High temperatures are terminal to wheat yield specifically in the flowering and grain filling stages during the second half of the growing season (Farooq et al., 2011). Increasing temperature change and heat stress events in

the recent decades and their impacts on wheat crop growth processes are extensively studied (Asseng et al., 2015; Lobell et al., 2012; Farooq et al., 2011; Ortiz et al. 2008). Another environmental factor that has been widely studied is the impact of increasing $[CO_2]$. The resulting $CO_2$ fertilization effect is found to promote crop growth (Dubey et al., 2015). Apart from environmental factors, management practices including nitrogen fertilizer application and irrigation also significantly affect wheat production (Myers et al., 2017; Leaky et al., 2009; Luo et al., 2009). Because wheat is grown in the non-monsoon months, it is a high irrigation crop with almost 94% of the wheat fields in India equipped for irrigation (MAFW, 2017). Quantification of the impacts of land management practices on crop growth helps in understanding how croplands can be managed to improve production (Tack et al. 2017).

Even though India is the third largest wheat producer in the world, domestic production is barely sufficient to meet the country's demand for food and livestock feed (USDA, 2018). Data from different sources report a relatively poor yield of wheat in India as compared to other countries (FAOSTAT, 2019). Hence, there is an urgent need to address this yield gap by developing better land-management practices under different environmental conditions (Stratonovitch et al. 2015; Zhao et al. 2014; Luo et al., 2009). A key first step to achieve this goal is to understand the processes involved in interactions of the crop with its environment and the factors responsible for impacting crop growth.

Dynamic Growth Vegetation Models (DGVMs) are well-established tools to study global climate-vegetation systems. Implementation of crop-specific parameterization and processes in DGVMs provides us with a better framework to assess and represent the role of agriculture in climate-vegetation systems (Song et al., 2013; Bondeau et al., 2007). This helps in better estimation of biogeochemical and biogeophysical processes, improves the representation of feedback mechanisms as well as prediction of yield and production. Multiple process-based

models with crop-specific representations are being used recently (e.g., Lu et al., 2017;
Drewniak et al., 2013; Song et al., 2013; Lokupitiya et al., 2009; Bondaeu et al., 2007) instead
of standalone crop-models for this purpose.
This study explores the effects of environmental drivers and management practices on spring
wheat in India using the land model ISAM (Song et al., 2015 and 2013). The specific objectives
of this study are: (1) to implement a dynamic spring wheat growth module in ISAM, and (2) to
study the effect of environmental factors (elevated $[CO_2]$ and climate change, including
temperature and precipitation change) and land-management practices (irrigation and nitrogen
fertilizers) on production of spring wheat in India for the 1980-2016 period using ISAM. To
the best of our knowledge, this is the first study that evaluates the impacts of multiple
environmental factors and land management practices on spring wheat in India at a country
level by implementing spring wheat specific processes in a land-surface model.
**2. Methods**
**2.1 Study design**
The study is designed as follows. First, field data on crop physiology is collected at an
experimental spring wheat field site. Next, the spring wheat model is developed and
implemented in ISAM. The model is run at site-scale for calibration and evaluation with the
site data. Next, the model is run for the entire country driven by gridded driver data and
evaluated with country-scale wheat production data. Finally, numerical experiments are
conducted to estimate the effects of various environmental factors and land-management
practices on spring wheat production. Details of each step are described below.
**2.2 Site Data**
Field data on spring wheat growth is required to develop, calibrate and evaluate the spring
wheat model. Such data is not readily available in the public domain. Hence, a field campaign

is conducted during two growing seasons, 2014-15 and 2015-16. Leaf area index (LAI) is measured for 2014-15 and LAI and aboveground biomass at different growth stages are measured for the growing season 2015-16 at a wheat experimental site. The site is approximately 650 m$^2$ in area and is located at 28°40' N, 77°12' E in the Indian Agricultural Research Institute (IARI) campus in New Delhi, which is a subtropical, semi-arid region. The crop was sown on 18th November 2014 and 20th November 2015. It reached physiological maturity on 30th March in both years. The wheat field is irrigated with unlimited amount to ensure that the water stress to the crop is minimal. Mimicking local farming practices, whenever the soil is perceived to be dry, water is added till the top layers are near saturated. These led to 4 irrigation episodes in 2014-15 and 5 in 2015-16. Total nitrogen fertilizer of 120 kg N ha$^{-1}$ is being added to the crop in three batches of 60, 30 and 30 kg N ha$^{-1}$ in a span of 60 days from planting day.

The LAI is measured at the weekly interval with Li-Cor LAI-2000 plant canopy analyzer that measures gap fraction at five zenith angles using hemispherical images from a fisheye camera. LAI is estimated by comparing one above-canopy and three below-canopy measurements. The observed LAI is actually an average of multiple (at least five) LAI observations at different locations in each plot.

For measuring above ground biomass, plant samples from 50 cm row length are cut just above the soil surface. Then, different plant organs like leaves, stem, and spike (after anthesis) portions of plant sample are separated out. These are initially dried in the shade and later dried at 65°C in an oven for 72 hours till the weight stabilizes. Finally, the weight of dried plant samples were measured using an electric balance. To measure yield, two samples of mature wheat crops are harvested from 1 × 1 m$^2$ area in each plot and allowed to air dry. The total weight of grains and straw in each plot is recorded with the help of a spring balance. After

thrashing and winnowing by mechanical thrasher, grains are weighed to estimate grain yield and thousand-grain weight.

## 2.3 Model Description

### 2.3.1 Dynamic C3 crop model in ISAM

ISAM is a well-established land model that has been used for a wide range of applications (Gahlot et al. 2017; Song et al. 2016, 2015, 2013; Barman et al. 2014a, 2014b). ISAM simulates water, energy, carbon, and nitrogen fluxes at a one-hour time step with $0.5^{\circ} \times 0.5^{\circ}$ spatial resolution. ISAM has vegetation-specific growth processes for all major plant-functional types implemented in the model to better capture seasonality for each. Song et al. (2013) have developed a soybean and maize model for ISAM. Because soybean and wheat are both C3 crops, the dynamic C3 crop model framework from the soybean model is used as a foundation to build a spring-wheat model for this study. The model structure, phenological stages, carbon and nitrogen allocation processes, parameters and performance are extensively described and evaluated in various studies (Song et al., 2016, 2015, 2013).

### 2.3.2 Development and implementation of spring wheat processes in ISAM

The spring wheat processes in ISAM are implemented using the C3 crop framework (Song et al., 2013). For this purpose, C3 crop specific equations and parameters are updated based on literature. The model equations are available in Song et al. (2013). A brief description is given in the online supplement and the revised parameters are available in Table S1. Some of the parameter values are collected from literature while the rest are estimated during model calibration.

ISAM accounts for dynamical planting (Song et al., 2013). This unique feature of ISAM is quite important for modelling wheat in India because in India wheat is grown in different

climatic conditions (Ortiz et al., 2008) and in multiple cropping systems. In the rain-dependent, tropical central parts of India, wheat is planted early; in eastern parts of India where rice is harvested before the wheat is planted on the same field, wheat is planted late; and it is timely sown in the northern and western parts of India (Table S2). ISAM uses different conditions based on a 7 day average of air temperature and 30 day total precipitation to dynamically calculate the planting day. Observed wheat planting and harvest dates (Sacks et al., 2010) are used to calibrate the planting time and harvest time criteria in the model along with other state-level and regional datasets (NFSM). This allows for correct simulation of the observed spatial variability of the planting date.

The heat stress effect is implemented to account for the observed negative effects of high temperatures on grains (Asseng et al., 2015; Farooq et al., 2011) during the reproductive stage of the phenology (Zhao et al., 2007). To include these effects, net carbon available for allocation to grains decreases as daily average temperatures increase from $25^{\circ}$ C to $35^{\circ}$ C in the flowering and grain filling stages (Table S3, Eq. A1-A3). This limits the growth of a plant. Beyond daily average temperatures of $35^{\circ}$ C, the grains fail to develop.

## 2.4 Site-scale simulations for calibration and validation

The spring wheat model is calibrated at site level using LAI and aboveground biomass data collected at the IARI site for the 2015-16 growing season using the protocol described in Song et al. (2013) and validated using LAI data for the 2014-15 growing season. ISAM can be configured to run for a single point. Using this capability, ISAM is run at site-scale to simulate spring wheat growth observed at the IARI site. The model is spun-up by recycling the Climate Research Unit-National Centre for Environment Prediction reanalysis data (CRU-NCEP, Vivoy et al., 2018), Global Carbon Project Budget 2017 [$CO_2$] (Le Quere et al., 2018) and airborne nitrogen deposition (Dentener, 2006) data for 2015-16 until the soil temperature, soil

moisture, the soil carbon pool and the soil nitrogen pool reaches a steady state. Then, the above
ground biomass carbon (leaves + stem + grain) is calibrated using aboveground biomass (Fig.
1a), nitrogen fertilizer amount added, sowing date and harvest date for the 2015-16 growing
season. Next, phenology-dependent carbon allocation fractions for leaves, stem, and grain are
calibrated, using the LAI data (Fig. 1b), duration, and heat unit index requirement for each
growth stage. The model is evaluated by comparing simulated and observed LAI for the 2014-
15 growing season.

**2.5 Gridded Data for country-scale simulations**
Driver data for environmental and anthropogenic forcings are required to conduct ISAM
simulations. ISAM is driven by $0.5^{o}X0.5^{o}$ climate data from CRU-NCEP (Viovy et al., 2018)
with 6 hourly mean surface air temperature, specific humidity, incoming shortwave and long-
wave radiations, wind speed and precipitation that are interpolated to hourly values. Annual
[$CO_2$] data is taken from the Global Carbon Project Budget 2017 (Le Quéré et al., 2018).
Spatially explicit annual nitrogen fertilizer data for wheat from 1901-2005 is created by
combining nitrogen fertilizer data from Ren et al., (2018) and Mueller et al. (2012) (Table S3:
Eq. A4-A5).

Gridded data for the wheat harvested area, nitrogen fertilizer application, and irrigation are
required as model input to estimate actual wheat production for India in recent years (1980-
2016). For this purpose, an annual spatially-explicit gridded wheat harvested area dataset for
India is created as a part of this study by combining spatially-explicit wheat area from
Monfreda et al. (2008) for the mean value over the time-period 1997-2003 (ca 2000) and non-
gridded state-specific annual wheat harvested area from the Directorate of Economics and
Statistics, Ministry of Agriculture And Farmers Welfare, India (MAFW, 2017) (Eq. A6, A7,
A8). Annual Area Equipped for Irrigation (AEI) dataset is created by linear interpolation of
decadal data from Siebert et al. (2015) (Eq. A9).

**2.6 Country-scale simulations**
Country-scale simulations are conducted after model calibration and evaluation. First, we spin
up the model for the year 1901 by repeating the climate forcing data of CRU-NCEP (Viovy,
2018) for the period 1901-1920, and fixed year (1901) data for $[CO_2]$ of 296.8 ppm and data
for airborne nitrogen deposition (Dentener, 2006), and zero amount of nitrogen fertilizer and
irrigation, until the soil temperature, soil moisture and the soil carbon and nitrogen pools reach
a steady state at approximately 1901 levels. Details of the spin-up process are described in
detail in Gahlot et al. (2017).  After the model spin-up, numerical experiments are conducted
as transient runs from 1901 to 2016. To estimate the effects of external forcings, country-scale
runs are conducted over wheat-growing regions in India by varying different input forcings
(Table 1). Control run ($S_{CON}$) represents the model run from 1901 to 2016 with time-varying
annual $[CO_2]$, climate data, annual grid-specific nitrogen fertilizer, and full irrigation to fulfil
the water needs of the crop. Four additional simulations are conducted by assigning a constant
value to each input forcing one at a time. For instance, in $S_{CO2}$, all input variables (temperature,
nitrogen, and irrigation) are the same as in the $S_{CON}$ case except $[CO_2]$ that is held constant at
1901 level. The difference in model simulations from $S_{CON}$ and $S_{CO2}$ then gives the effect of
elevated $[CO_2]$ on wheat crop growth processes. Here we present the results only for the recent
decades, 1980 to 2016.

Model performance at the country-scale is evaluated by comparing the model simulated total
wheat production at the country level with FAOSTAT (2019) and the Directorate of Economics
and Statistics, Ministry of Agriculture and Farmers Welfare (MAFW, 2017) data. The

production for each grid cell is an area-weighted sum of production from irrigated and rainfed area fractions (Equation A10).

To study the spatial variation in production, the wheat-growing regions of India are divided into spring wheat environments (SWE) based on the mega-environment concept (Chowdhury et al., 2019). For this purpose, we divide the wheat-growing regions of India into five SWEs (Fig 2) based on temperature, precipitation, and area equipped for irrigation (Table 2) to identify regions with similar growing conditions for wheat. SWE 1 (Fig. 2) represents mostly the Indo-Gangetic plains that offer good access to irrigation for wheat which is a non-monsoon crop. The growing season temperatures fall in the optimum range for wheat growth. SWE 2, which mainly comprises of the wheat growing regions in the proximity of the Himalayas, is characterized by very low growing season temperatures and high rainfall. SWE 3 represents the north-western parts of the country with moderate to high growing season temperatures, low rainfall and small values of AEI. SWE 4 represents the central parts of India and tropical wheat growing regions with high temperatures and moderate growing season precipitation. SWE 5 represents the crucial wheat growing regions of the country where the conditions are similar to SWE 1 but irrigation facilities are lacking. Wheat production for each of the SWEs has been discussed further in the following sections.

## 3. Results

### 3.1 Spring wheat model evaluation

The simulated magnitude and intra-seasonal variability in LAI for 2014-2016 compared well with the experimental wheat site at IARI, New Delhi (Fig. 1c).

Spatial distribution of model estimated wheat production at a country scale is compared well, including the highly productive Indo-Gangetic plains, with the data from Monfreda et al. (2008)

for the year 2000 (Fig. 3). ISAM simulated country scale wheat production for 1980-2014 also
compares well with production data from FAOSTAT (2019) and MAFW (2017) datasets (Fig.
4) with correlation coefficients of 0.92 and 0.91 respectively with the two datasets. However,
the model estimated production is slightly higher than both observed datasets. This may be
attributed to the fact that the model is calibrated to the high-yielding wheat cultivars grown in
recent years (2015-16). Hence, the model is a valid tool to study interactions of wheat with its
environment for recent years.

**3.2 Effects of environmental and anthropogenic forcings at country scale**
In this study, we examine the effects of two environmental factors ($[CO_2]$ and temperature
change) and two land management practices (nitrogen fertilizer and water available) on the
production of spring wheat. The impact of these factors is quantified as the difference between
the control and the experimental simulations (Eq. A11) described in Table 1. Results show that
during the 1980-2016 period, $[CO_2]$, nitrogen fertilizers and water available through irrigation
have a positive impact on wheat production but the impact of temperature is negative (Fig. 5)
due to reasons detailed below. The effects of $[CO_2]$, temperature change, addition of nitrogen
fertilizers and irrigation show a trend of 0.67, -0.39, 0.26 and 0.25 Mt yr$^{-1}$ over the period 1980-
2016, respectively (Table 3).

$CO_2$ fertilization is the most dominant factor that has contributed to increase in wheat
production over India. Annual average $[CO_2]$ worldwide has increased from 337.7 ppm in 1980
to 404.3 ppm in 2016. This increase in levels of $[CO_2]$ at the rate of 1.82 ppm yr$^{-1}$ has promoted
growth in wheat as elevated $[CO_2]$ levels are known to enhance photosynthetic $CO_2$ fixation
and have a positive impact on most C3 plants (Myers et al. 2017; Leakey et al. 2009; Allen et
al., 1996). Our results show that for every ppm rise in $[CO_2]$ level total wheat production of the
country has increased by 0.37 Mt (Fig. 6a; Table 3). This amounts to a 22 Mt (30%) increase

in production compared to the 1980-84 period due to increased $[CO_2]$ levels. A positive correlation coefficient of 0.97 between annual wheat production and annual $CO_2$ concentration confirms a positive impact of $[CO_2]$ on wheat production. Other studies based on multiple approaches including experiments have also shown an increase in yield and growth of C3 crops under high $[CO_2]$ conditions (Dubey et al., 2015; Leakey et al., 2009).

Nitrogen fertilizers are added to the farmland to reduce nutrient stress to the crop. The use of nitrogen fertilizers is important in the Indian context due to two reasons. First, India is a tropical country where higher temperatures and precipitation cause loss of nitrogen from the soil due to denitrification. Second, crop nitrogen demand is high because multiple cropping is widely practiced. The average amount of nitrogen fertilizer added per unit area shows a positive trend of 2.71 kg N ha yr$^{-1}$ during 1980-2016. This implies an increase in total wheat production at the rate of 0.10Mt for every kg N ha added to the farm (Fig. 6c; Table 3). This amounts to an 10.63 Mt (15%) increase in production compared to the 1980-84 period due to increased fertilizer application.

Irrigation is a key factor for spring wheat in India where 93.6% of the wheat area is equipped for irrigation (MAFW 2017), most of irrigated area being concentrated in the Indo-Gangetic Plains. Unfortunately, data on the actual amount of water used for irrigation water is not available. Hence, in the $S_{CON}$ simulation, we consider every grid cell is 100% irrigated so that the crops do not undergo water stress at any point in the growing season. This is to say that irrigation water required in the model is dependent on water demand of the crop. With this condition, our results show that with all the regions 100% irrigated, wheat production shows a positive trend during 1980-2016. Overall, there is a 8.47 Mt (12%) increase in production compared to the 1980-84 period due to increased irrigation.

The average air temperature for the wheat-growing season months (October-March) during the
study period (1980 to 2016) has shown an increase at the rate of 0.026 $^{\circ}$C yr$^{-1}$. Higher
temperature during second half of the growing season is specifically known to produce smaller
grains and low grain numbers (Stratonovitch et al., 2015; Deryng et al., 2014). Our results have
shown a decrease of 8.38 Mt (~10% reduction) of wheat per degree Celsius increase in average
growing season temperature (Fig. 6b). This is higher than the global estimate of 6% reduction
per degree Celsius rise in mean temperature (Asseng et al., 2015). Studies have reported that
wheat-growing regions in low-latitudes are more susceptible to rising temperatures (Tack et
al., 2017; Rosenzweig et al., 2014) since optimum temperatures in these regions have already
been reached. Overall, there is a 13 Mt (18%) reduction in production compared to the 1980-
84 period due to rise in average growing season temperatures.

In the presence of all input forcings (S$_{CON}$), the trend of wheat production in India remains
positive at 1.17 Mt year$^{-1}$ from 1980 to 2016.

**3.3 Effect of environmental and anthropogenic forcings at the regional scale**
It is clear that environmental and management factors significantly affect wheat production at
a country scale. It is important to understand how these factors can affect production for
different regions. For this purpose, the results of the control simulation (S$_{CON}$) with all the
forcings are analysed for each of the SWEs shown in Fig. 2. A SWE is representative of similar
climatic and environmental conditions regionally in which wheat is grown. One SWE differs
from the other in terms of different temperature range, precipitation received and irrigation
availability. The S$_{CON}$ case is analysed to ensure that the input factors are fully implemented in
the model-estimated production and their effect can be studied effectively. One important thing
to note is that irrigation in the model is calculated as the excess water demand required by the
crop to grow in no-water-stress conditions. Hence, the S$_{CON}$ calculates irrigation as the ideal

case scenario assuming that all the water demand of the crop is met. Overall, this analysis will identify the factors (environmental conditions and land management practices) that predominantly drive the wheat production range in a given SWE.

The results of this regional analysis are presented in Fig. 7 showing scatterplots of production as a function of various drivers for each wheat-growing grid cell in the model. A similar plot showing the relationship between production, AEI and wheat area is presented in Fig. 8. Together, these two figures allow us to understand how different environmental factors and management practices can affect production in different SWEs. Atmospheric $[CO_2]$ is omitted from this analysis because it is assumed to be spatially uniform.

The Indo-Gangetic plain region (SWE1) is the best-suited environment for growing spring wheat in India due to favourable growing season temperatures (Fig. 7a), moderate to high fertilizer application (Fig. 7b), high availability of irrigation facilities (Fig. 8b), and moderate water demand (Fig. 7c). Hence, SWE1 is the major contributor to the annual total wheat production of India. Low temperatures (Fig. 7a) in the Himalayan foothills region (SWE2) result in the limited production of wheat in this region. High rainfall in growing season months is helpful and hence, limited amount of water is required for irrigation (Fig. 7c) in this area. The arid north-western India region (SWE3) is very low in production due to the high temperatures (Fig. 7a) coupled with lack of irrigation facilities (Fig. 8b) needed to mitigate the high water demand created by low precipitation. SWE4 in the central and north-eastern India is also low in production due to high temperatures during growing season (Fig. 7a) even though the water demand is low (Fig. 7c) due to moderate rainfall. SWE5 areas in the south-central India have limited wheat production because of limited irrigation facilities (Fig. 8b) despite favourable temperature conditions.

Wheat production is directly proportional to area on which wheat is cultivated in a given region/SWE (Fig. 8a). Fig 8b shows that wheat production is, in fact, positively correlated to AEI at the grid level. Since production in this analysis is derived from the $S_{CON}$ case and no AEI data is used in its calculation, it is interesting to see such a strong correlation between wheat production and AEI at grid level that are two independent datasets. This can be explained by Fig. 8c that clearly indicates that availability of irrigation (high AEI) is a major factor that drives area on which wheat is cultivated in a grid cell. Wheat, being a non-monsoon crop, is highly dependent on availability of irrigation in a region. For regions with high growing season temperatures, additional water stress is induced in the crop along with heat stress that limits crop production. Hence, availability of favourable temperatures is crucial to ideal growing conditions for wheat. If irrigation can be made available in these regions, like in SWE 5, wheat cultivation area and wheat production can significantly grow in the years to come.

Similar to the analysis done for country-scale impact of different factors, we quantified the impact of factors on different SWEs. The results of this analysis are summarized in Table 4. SWE1 and SWE5 are the two regions where magnitude of trends in change in wheat production with different input forcings are the highest (Table 4). The magnitudes of impacts of forcings on SWEs 2, 3 and 4 are relatively small. This is because the analysis involves production that is calculated as yield times the harvested area. The numbers in Table 4, hence, do not reflect changes per unit harvested area.

While $CO_2$ fertilization, water added through irrigation and nitrogen fertilizers are found to increase wheat production in SWE1 at 0.26 Mt per ppm [$CO_2$], 0.35Mt per 1000 mm and 0.07 Mt per kg N ha$^{-1}$ respectively, production is found to decrease by 3.52 Mt for every degree Celsius rise in average growing season temperatures. It is found that water added through irrigation has small yet negative impact on production in SWE2. This can be due to excess

surface runoff in SWE2 that might lead to washing away of nitrogen from the soil resulting in nutrient stress in the crop. The impact of different forcings is also found to be significant for SWE5 where $[CO_2]$, irrigation and nitrogen fertilizers have promoted wheat production at the rates of 0.07 Mt per ppm $[CO_2]$, 0.41 Mt per 1000 mm and 0.01 Mt per kg N ha$^{-1}$, respectively. Irrigation is seen to have the most impact on wheat production in SWE 5 out of all the SWEs.

**4. Conclusions and Discussions**

This study explores the effects of environmental drivers and management practices on spring wheat in India using the land model ISAM. For this purpose, we build a dynamic spring wheat growth processes for ISAM where (i) we parameterize and calibrate the equations in the C3 crop model framework available in ISAM, (ii) develop new equations for dynamic planting time and heat stress, (iii) collect field data to calibrate and evaluate the model at site scale and (iv) develop gridded datasets of wheat cultivated area, irrigation and nitrogen fertilizer data to conduct country-scale simulations. The model is able to simulate the spatio-temporal pattern of spring wheat production at the country-scale. This evaluation implies that the model can serve as a simulation tool to conduct numerical experiments to understand the behaviour of spring wheat.

In order to quantitatively study the role of environmental and anthropogenic factors, we conducted a series of numerical experiments by switching off one factor at a time. Our analysis focuses on the 1980-2016 period. Results show that the increase in $[CO_2]$ has a positive impact on wheat production due to the $CO_2$ fertilization effect. Atmospheric CO2 concentration has increased at 1.82 ppm yr$^{-1}$ and production has increased at a rate of 0.37 Mt per ppm rise in $[CO_2]$ since the 1980s that translates to a 22 Mt (30%) increase in countrywide production during the study period. This is consistent with observational studies such as Kimball (2016) that show an increase in yield of C3 grain crops due to elevated $[CO_2]$.

413

Application of nitrogen fertilizer has increased at 2.71 kg N ha$^{-1}$ yr$^{-1}$ leading to increased production of spring wheat at the rate of 0.10 Mt for every kg N ha$^{-1}$ added that is equivalent to a 10.63 Mt (15%) increase in countrywide production during the study period. Nitrogen deficiency is very high in India because of high consumption due to multiple cropping and nitrogen loss due to denitrification of the soil aided by the tropical climate. Nitrogen fertilizer contributes to increased production by mitigating this nutrient deficiency.

Our model results suggest irrigation increase could have led to an increase in production of spring wheat at a rate of 0.31 Mt per 1000 mm of water added implying a 8.47 Mt (12%) increase in countrywide production during the study period. Irrigation appears to be the most important factor controlling production across all the spring wheat environments. We note here that in our experiments irrigation is equivalent to 'no water stress'. This approach seems to be the best option because data on actual water use in irrigation is not available. In grid cells that are equipped for irrigation, we set the water stress term to zero. In reality, water stress may not go to zero in some areas where water or power availability is limited. In these areas, the model underestimates the simulated effect of irrigation on productivity.

Average growing season temperatures have increased by 0.026 °C yr$^{-1}$ leading to a productivity loss of 8.38 Mt (~10%) per degree Celsius rise in temperature that is equivalent to a 13 Mt (18%) decrease in countrywide production during the study period. Crop heat stress is a major reason behind this loss. The optimum temperature for wheat is 25 °C in the reproductive stage. Heat stress effect triggers in the model when the canopy air temperature higher than 25 °C and lesser than 35 °C reduce grain filling and negatively impact the growth of storage organs. The observed 10% reduction rate in production is higher than the global average of 6% (Asseng et

al. 2015) because the growing season temperatures in India are already near the upper limit of the optimal range.

The regional-scale analysis shows that the SWE1 is the best environment for growing spring wheat in India due to favorable growing season temperatures, moderate water demand, and availability of irrigation facilities. Hence, this region is the main contributor to the annual total wheat production of India. Northwestern India (SWE3) covering the states of Rajasthan and Gujarat is the least productive region due to high growing season temperatures coupled with a lack of irrigation facilities needed to mitigate the high water demand created by low precipitation. Studies have concluded that in order to improve and represent crop growth processes in the models and to increase certainty in model-based assessments, there is a need for more focused regional-scale studies (Maiorano et al. 2017; Koehler et al. 2013). This study is an attempt to work in similar direction with focus on wheat in India.

Apart from advancing our understanding of spring wheat growth processes, the crop model can also contribute to real-world decision-making. For example, our results show that wheat production in India has steadily increased at a rate of 1.17 Mt/year from 1980 to 2016. This implies that the negative effect of rising temperatures was offset by positive contributions from other drivers. Our model can be used to conduct experiments to identify optimal solutions to future scenarios. Furthermore, using crop-specific models like the spring wheat model developed in this study will improve the simulation of crop phenology for agro-ecosystems. This will likely lead to better estimates of carbon fluxes and their spatio-temporal variability.

The Earth System is a nonlinear system where different components interact with each other. In this study we used a process-based model that includes such interactions including interactions and feedbacks between different drivers. For instance, higher temperatures

increase the crop water demand. Higher $[CO_2]$ increases photosynthesis that also affects nutrient and water demand. Because of these interactions, the sum of the effects will not add up to 100%. Moreover, the experiments conducted in this study are not exhaustive; there are other factors like relative humidity, solar radiation etc. that might affect production.

There is scope for improving the crop model and the modelling framework. The processes involved in $CO_2$ fertilization need improvement to match the FACE studies. The addition of new processes accounting for the effects of pests and multiple cropping will make the simulations more representative of the Indian situation. Better data will also improve the fidelity of the simulations. A key bottleneck in simulating crop growth at regional-to-global scales is the lack of irrigation water use datasets. To the best of our knowledge, large-scale observation-based datasets of water used in irrigation do not exist even though there are numerous datasets for irrigated areas and areas equipped for irrigation (e.g., Zohaib et al., 2019). The development of irrigation water use datasets will reduce the uncertainty in simulating the effect of water stress on crop production. Equipped with these improvements, ISAM can become an indispensable tool for informing policy on food security and climate change adaptation.

**Code and data availability**

ISAM model code is available upon request.

**Electronic supplement**

Electronic supplement has been submitted separately.

**Author contribution**

SG, AKJ and SBR conceptualized the study; SG, TSL and AKJ designed the numerical experiments and generated the input datasets; SG conducted the numerical experiments and analyzed the outputs; VKS and RD collected the field observations; SG, AKJ and SBR wrote the paper.

**Competing interests**

The authors declare no competing interests.

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

     **Figures and Tables**

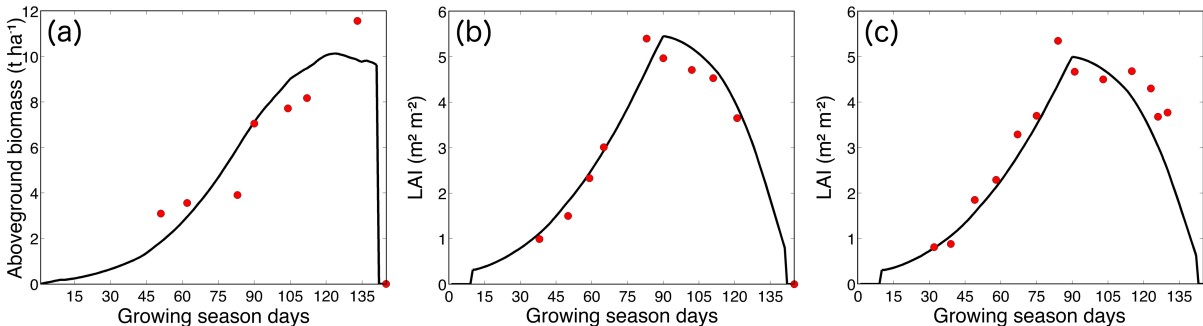

Figure 1: Model calibration and validation plots for the experimental wheat site at IARI, New Delhi. (a) Model calibration for aboveground biomass for growing season 2015-16. (b) Model calibration for LAI for growing season 2015-16. (c) The model estimated LAI validated with site-measured data for growing season 2014-15. The red dots are site-measured values and the black lines are ISAM simulated values.

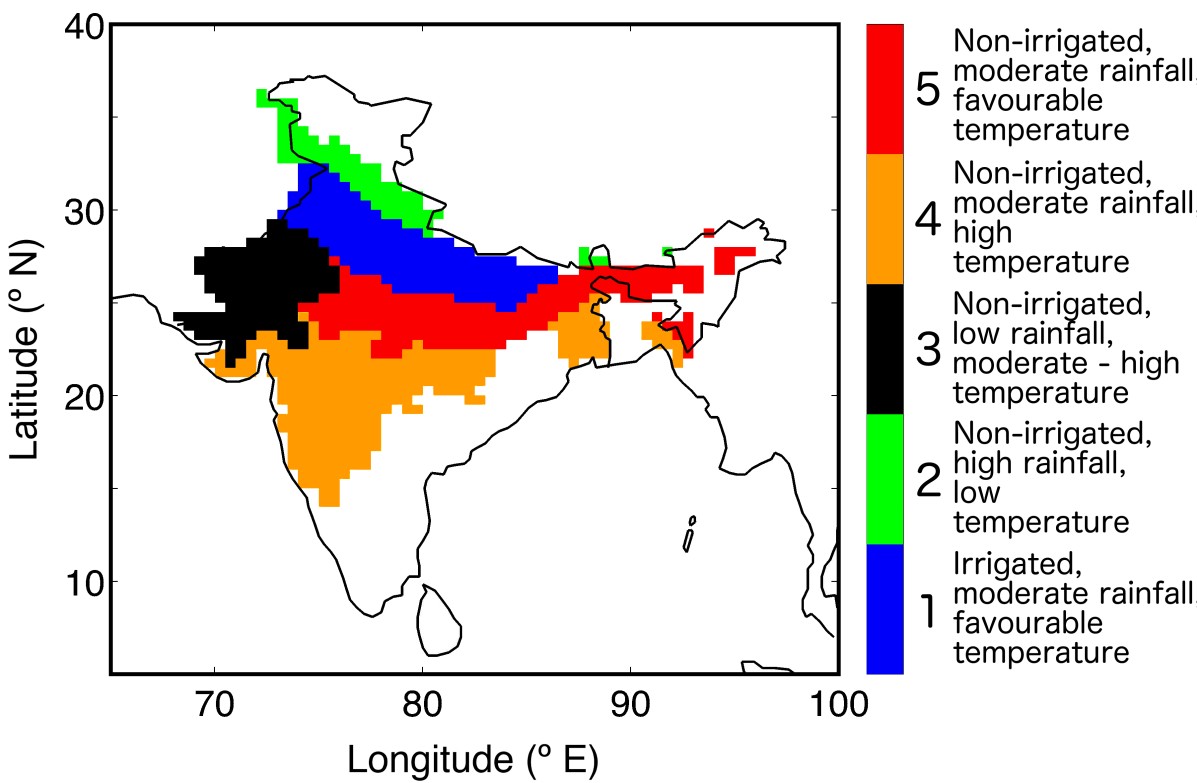



Figure 2: Classification of wheat growing areas into spring wheat environments (SWE) in
India.

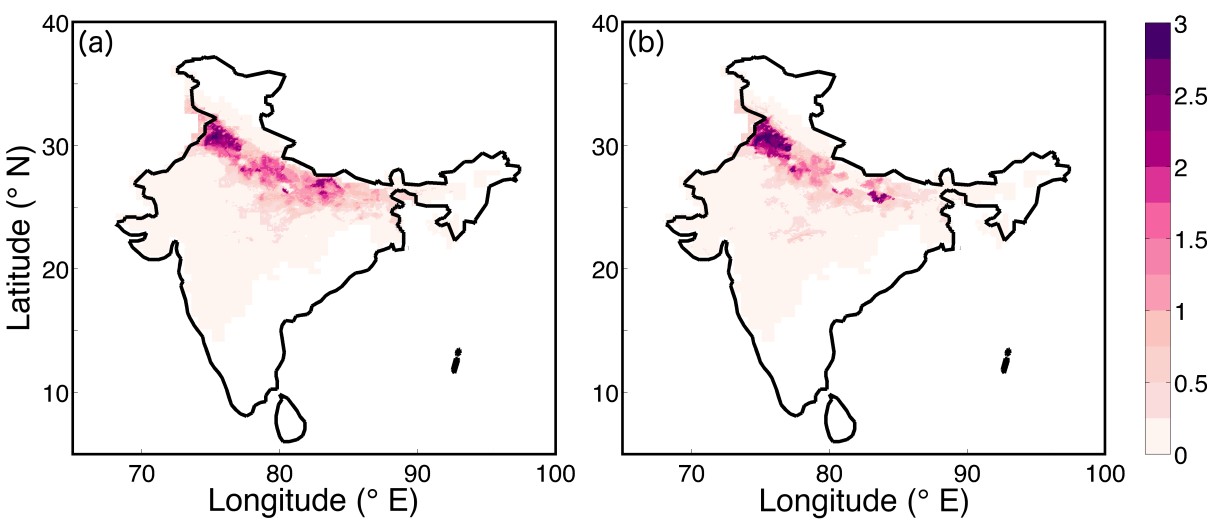

Figure 3: Wheat production (X 10⁴ tonnes) averaged for 1997-2003 (a) simulated by ISAM and (b) observed M3 dataset (Monfreda et al., 2008).

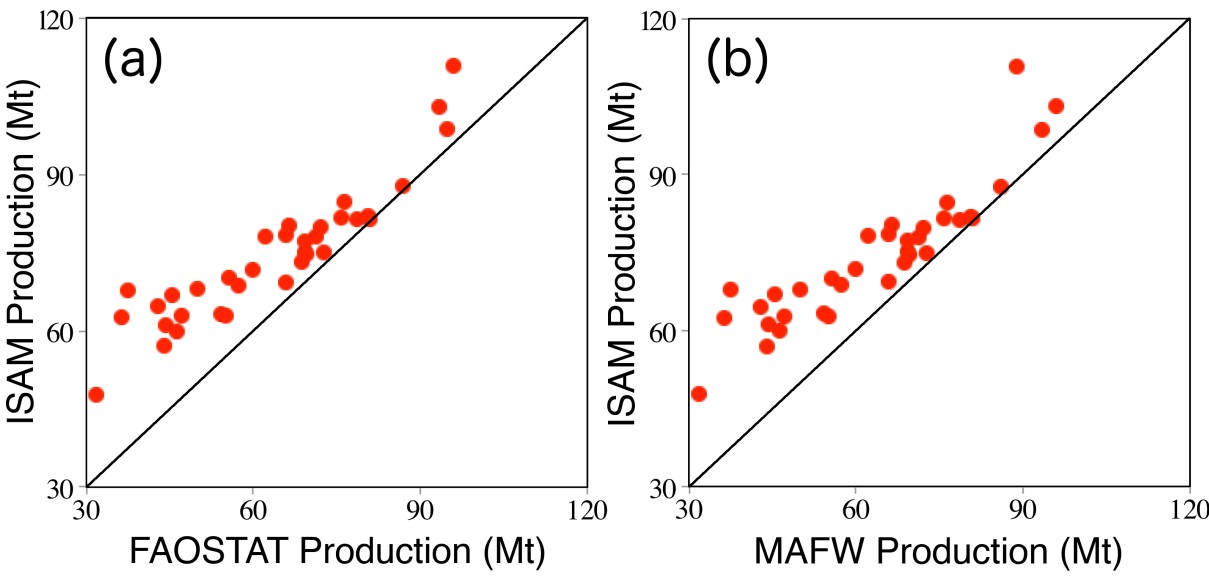

Figure 4: Scatter plots of the ISAM simulated wheat production (Mt) compared to (a) FAOSTAT (2019) and (b) the Directorate of Economics and Statistics, Ministry of Agriculture and Farmers Welfare, India (MAFW, 2017) datasets from 1980 to 2014. The Pearson's correlation coefficients are (a) 0.92 and (b) 0.91.

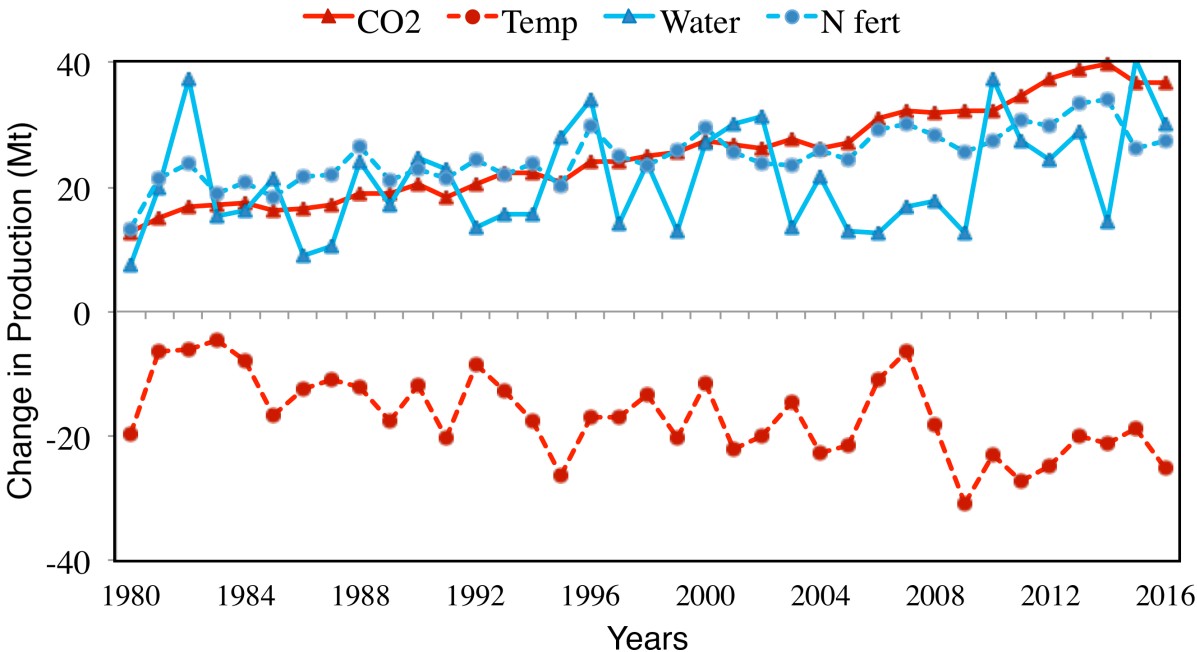

Figure 5: Impact ($S_{CON}$-$S_{<factor>}$) of different environmental factors (atmospheric $CO_2$ and changing temperature) and land management practices (nitrogen fertilizer and water availability) on production for 1980 to 2016.

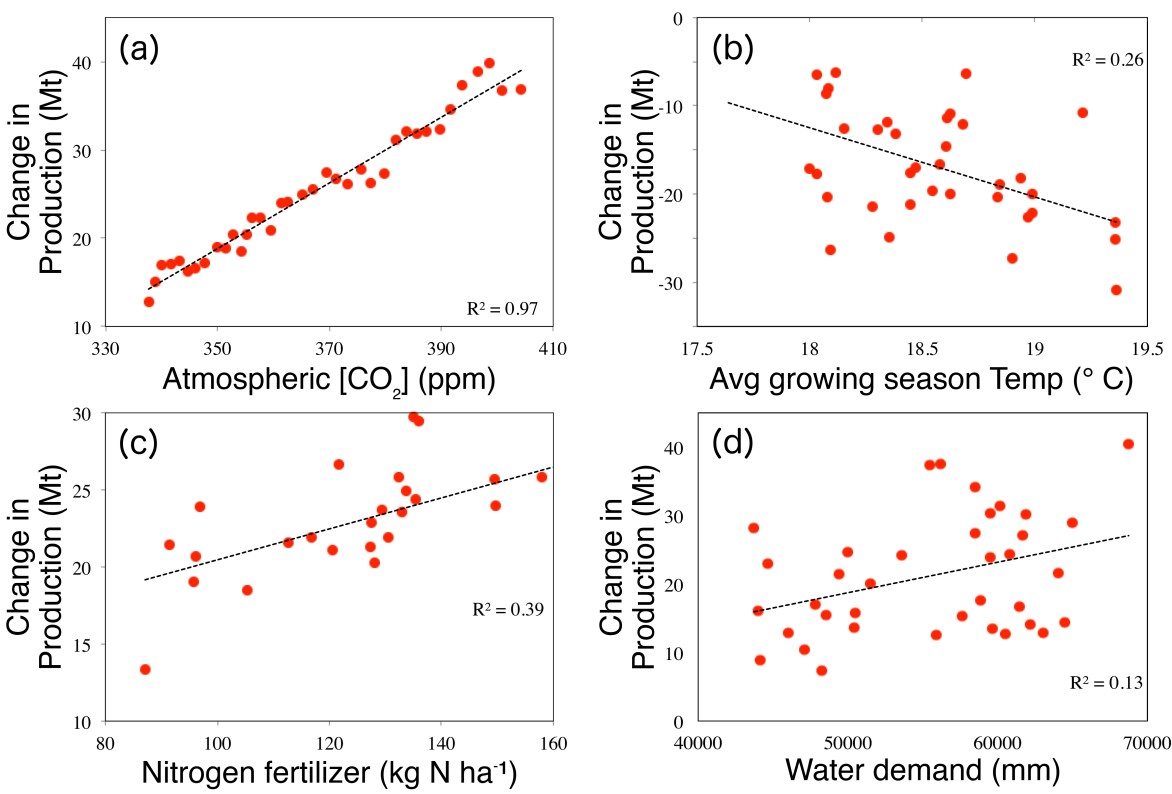



Figure 6: Plots of change in annual wheat production from 1980 to 2016 ($S_{CON}$-$S_{<factor>}$) with
annual (a) atmospheric $CO_2$, (b) average growing season temperature, (c) average nitrogen
fertilizer and (d) water demand. The black line shows Sen's slope (Sen, 1968).

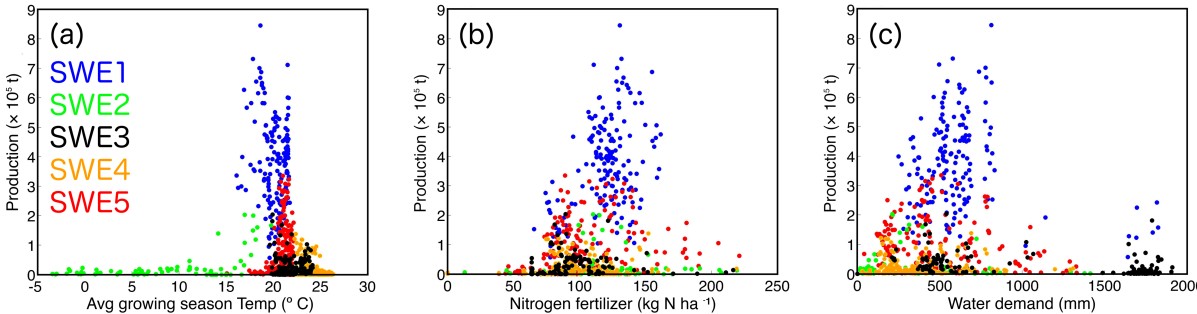



Figure 7: Scatter plots of grid-specific average wheat production from 1980 to 2016 with temporal average of input forcings (a) growing season temperature (b) nitrogen fertilizer and (c) water demand for different SWEs.

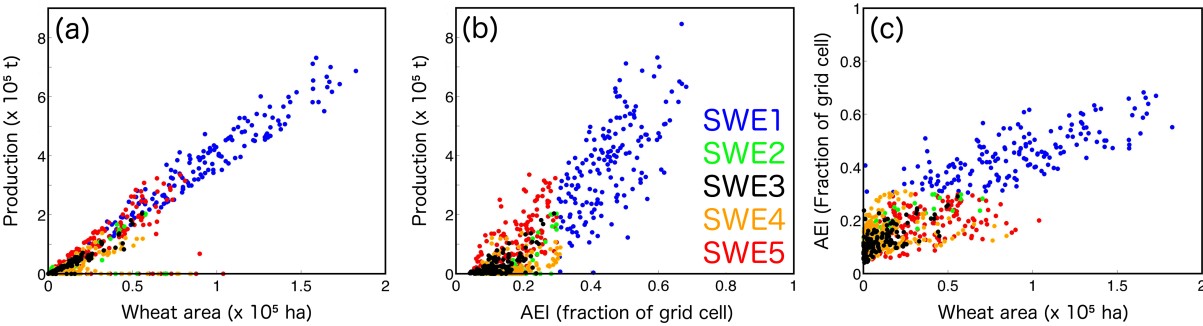

Figure 8: Scatter plots for gridded wheat production with the wheat area and Area Equipped for Irrigation (AEI) for different SWEs.

**Table 1: Description of numerical experiments conducted with ISAM wheat model from**
**1901 to 2016.**

| Numerical Experiments | [CO$_2$] | Temperature | Nitrogen fertilizers | Irrigation |
|---|---|---|---|---|
| **Control (S$_{CON}$)** | Annual values from Global Carbon Project Budget 2017 | 6 hourly CRU-NCEP | Grid-cell specific fertilizer amount (Source: this study) | Hourly values to ensure no water stress |
| **S$_{CO2}$** | **Fixed at 1901 level** | Same as in CTRL | Same as in CTRL | Same as in CTRL |
| **S$_{TEMP}$** | Same as in CTRL | **No temperature change*** | Same as in CTRL | Same as in CTRL |
| **S$_{N\_FERT}$** | Same as in CTRL | Same as in CTRL | **No fertilizer** | Same as in CTRL |
| **S$_{WATER}$** | Same as in CTRL | Same as in CTRL | Same as in CTRL | **No Irrigation + No precipitation change*** |
| **S$_{IRRI}$** | Same as in CTRL | Same as in CTRL | Same as in CTRL | **No Irrigation** |


*Data for years 1901-1930 is recycled to represent stable (no change) conditions

**Table 2: Characteristics of different spring wheat environments (SWE) in India.**

| Spring Wheat Environment (SWE) | Description | Geographic location | Average growing season temperature (ºC) | Average Growing Season Precipitation (mm) | Fraction of grid Area Equipped for Irrigation (AEI) |
|---|---|---|---|---|---|
| SWE1 | Irrigated, moderate rainfall, favourable temperature | Indo-Gangetic Plains | 17-22 | 30-150 | >=30% |
| SWE2 | Non-irrigated, high rainfall, low temperature | Himalayan Belt | <18 | >120 | <30% |
| SWE3 | Non-irrigated, low rainfall, moderate to high temperature | North-west India | 19-24 | <42 | <30% |
| SWE4 | Non-irrigated, moderate rainfall, high temperature | Central and southern parts of India | >21 | >40 | <30% |
| SWE5 | Non-irrigated, moderate rainfall, favourable temperature | Central parts of India | 17-22 | >40 | <30% |

**Table 3: Temporal variations of different input forcings and their impacts on annual**
**wheat production in India during the study period (1980-2016).**

| Input Forcing ($i$) | Rate of change of $i$ in study period | Rate of change in annual wheat production | Change in annual wheat production per unit change in $i$ |
|---|---|---|---|
| Elevated atmospheric $CO_2$ level | 1.82 ppm yr$^{-1}$ [a] | 0.67 Mt yr$^{-1}$ [a] | 0.37 Mt ppm$^{-1}$ [a] |
| Average growing season temperature* | 0.026 °C yr$^{-1}$ [a] | -0.39 Mt yr$^{-1}$ [a] | -8.38 Mt °C$^{-1}$ [a] |
| Average water demand | 443.94 mm yr$^{-1}$ [a] | 0.25 Mt yr$^{-1}$ [b] | 0.31 Mt 1000 mm$^{-1}$ [b] |
| Average nitrogen fertilizer per unit area | 2.71 kg N ha yr$^{-1}$ [a] ** | 0.26 Mt yr$^{-1}$ [a] | 0.10 Mt kg N ha$^{-1}$ [a] |


*October to March
**Data available from 1980-2005
[a] Trends are significant at $p < 0.01$
[b] Trends are significant at $p < 0.1$

**Table 4: Impacts of different external forcings on annual wheat production in the SWEs during the study period (1980-2016).**

| Input Forcing ($i$) | Change in annual wheat production per unit change in $i$ | | | | |
|---|---|---|---|---|---|
| | SWE1 | SWE2 | SWE3 | SWE4 | SWE5 |
| Elevated atmospheric $CO_2$ level (MT ppm$^{-1}$) | 0.26[a] | 0.02[a] | 0.01[a] | 0.02[a] | 0.07[a] |
| Average growing season temperature* (Mt ° C$^{-1}$) | -3.52[b] | -0.03 | -0.12 | -0.36 | -1.36 |
| Water demand (Mt 1000 mm $^{-1}$) | 0.35[b] | 0.04[b] | 0.61[a] | 0.07 | 0.41 |
| Average nitrogen fertilizer per unit area (Mt kg N$^{-1}$ ha$^{-1}$) | 0.07[a] | 0.01 | 0 | 0 | 0.01[b] |

[a] Values are significant at 99%

[b] Values are significant at 90%