# Peer review of "Impact of environmental changes and land-management practices on wheat production in India"

_Earth System Dynamics, 2020_

## Referee Comment (RC1) · Anonymous Referee #1 · 21 Apr 2020

Based on a dynamic land model, the authors developed the growth processes for spring wheat using field experiments and studied the effects of different environmental factors and land management practices on the spring wheat production in India. The authors have shown that both the increase in CO2 concentration, availability of water through irrigation and additional nitrogen fertilizer enhance the annual wheat production, while the elevated temperature reduces total wheat productions. The authors also investigate the impact of the above factors on wheat production at five spring wheat environment (SWE) regions. The paper is written in a very decent way and the results contribute to our understandings of the impact of induvial environmental factors on wheat production in India.

I have only some minor points to make. First, in the Results section, you discussed

changes in the overall wheat production. It might be interesting to also include changes in its components, like gross production and respiration in a table. Second, in section 3.3, you analyzed the wheat productions in five SWE regions and their associations with different environmental factors based on the control simulation results. You can also do a similar calculation as done at the country-scale, by comparing results between the control simulation and four sensitivity simulations (that is, you set a constant value to each environmental factor one at a time). In this way, you can probably see the impact of different factors on wheat production at different regions and quantify their overall contributions to the country-scale changes.

---

## Referee Comment (RC2) · Anonymous Referee #2 · 13 May 2020

This study uses the land surface model, ISAM, to examine the effect of different environmental factors, including atmospheric CO2, temperature, nitrogen fertilization, and irrigation on spring wheat production in India. First, the authors implemented spring wheat processes in ISAM by updating C3 crop parameterizations. After calibrating and validating the updated model against available observations, ISAM is applied to explore environmental and land management factors on Indian wheat production. It is found that during the last 30 years, increasing atmospheric CO2, addition of nitrogen fertilizer, and irrigation act to increase the production of spring wheat, but increased growing season temperature causes a loss of wheat production due to increased heat stress. Regional scale analysis of environmental factors and land management practice shows that Indo-Gangetic plain is the best region for growing spring wheat in India,

and Northwestern India is the least productive region for wheat growth.

This study makes a useful contribution to boost our general understanding in the effect of environmental change and land management on crop yields and production. The manuscript is in general clearly written. I recommend its publication after the following issues are addressed.

Lines 61-63: This sentence is hard to read. Please rephrase.

Line 105: How big is the wheat experimental site?

Lines 139-140: What are these major plant functional types?

Line 179: Where 'the climate driver' data come from? Also, the reference of Meinshausen et al., 2011 is missing.

Line 225: Section 3.2 Here the effect of a single factor ($CO_2$, temperature, etc.) is obtained by subtracting the simulation that includes the effect of all factors (CTRL) from the simulation that excludes the effect of a certain factor. Thus, the effect includes interactions with other factors. How would it compare with the sole effect of a certain factor by keeping other factors constant? (For example, suppose a simulation in which only atmospheric $CO_2$ changes to represent the $CO_2$ effect). Some discussion on this issue would be helpful.

Also, what are the nonlinear interactions among different factors? Does the sum of individual effects add linearly to the combined effect? The authors stated that changes in atmospheric $CO_2$, irrigation, fertilizers, and temperature led to 39%, 15%, 20% and -16% changes in countrywide production. So, what explains the residual change in wheat production that are not attributed to these factors? Some discussions should be added.

Lines 256-257: '2' -> 'two'

Lines 431-432: This sentence lacks a context. How does this study imply that ISAM

will likely to provide better estimate of terrestrial carbon flux?

Table 1: For the experiment STEMP that assumes no temperature change, I assume other climate fields such as precipitation and humidity change with time. If so, to what extent changes in other climate fields such as precipitation and soil moisture contribute to the 'direct' heat stress effect? Some discussions should be provided.

Table 3: statistic test should be done on the trends shown here.

---

## Referee Comment (RC3) · Anonymous Referee #3 · 20 May 2020

Comments on the MS The manuscript entitled "Impact of environmental changes and land-management practices on wheat production in India" is a very good study to quantify the role of various environmental factors and agricultural management practices on spring wheat production in India during 1980-2016 by Integrated Science Assessment Model (ISAM). Elevated atmospheric $CO_2$ and rising temperature are considered in environmental factors while nitrogen fertilizers applications and water availability through irrigation practices are considered in land management factors. The author's effort is commendable however some minor corrections needed in the draft: 1) From 1980-2016 for every ppm rise in [$CO_2$] level total wheat production of the country has increased by 0.37 Mt and 39% increase in production compared to the 1980-84 period (Fig. 6a; $R^2$ 0.97 while described in the draft $R^2$ 0.93) and thus a strong positive corre-

lation has observed. While during the same period total wheat production increased at the rate of 0.10Mt for every kg nitrogen fertilizer-N/ha applied to the farm and 20% increase in production compared to the 1980-84 period (Fig. 6c; R2 0.39), a decrease of 8.38 Mt (∼10% reduction) of wheat per degree Celsius increase in the average growing season temperature (Fig. 6b; R2 0.26) and 15% increase in production compared to the 1980-84 period due to increased irrigation. (Fig. 6d; R2 0.13). These factors have not shown a strong correlation that needs to improve. 2) In the draft different equations of the models are not shown e.g. Eq. A1, A2, A3 (line no. 170), Eq. A4, A5 (197); Eq. A6, A7, A8 (205); Eq. A9 (207), Eq. 10 (233), Eq. 11 (259) or may be included in the additional/supplement materials. 3) Data for the actual amount of water used for irrigation is not available. So in the SCON simulation, every grid cell is considered 100% irrigated and crops do not undergo water stress at any point in the growing season and all the regions are 100% irrigated. Since wheat is a non-monsoon crop, is highly dependent on the availability of irrigation. The development of irrigation water use datasets could reduce the uncertainty in simulating the effect of water stress on crop production. 4) Variation in wheat productivity in different regions as well as in different years of the study period (1980-2016) depends not only on environmental factors and management practices but also on the genetic factors, multiple cropping's, insect pests and diseases. Since 1980 various hybrids and high yielding wheat varieties were cultivated to increase the input use efficiency and higher economic yield. Similarly, in different climatic zones, area-specific resistant wheat varieties were also grown to enhance wheat productivity. The addition of new processes accounting for the effects of pests, multiple cropping and genotypes will make the simulations more representative of the Indian situation. 5) The study is more generalized for different climatic zones/spring wheat environment (SWE) while there is a need for more focused regional-scale studies. However, the study is an attempt to work in the similar direction with a focus on wheat in India. 6) In the draft multiple citations should be arranged in descending order of the publication year (line no. 54, 58-59, 68-69, 76, 79-80, 83, 137, 145, 212, 270-271, 277-278, 303) and citation of line no. 76, 212, 213 needs to correct
as per the formatting guidelines of the journal. 7) References missing for some of the citations in the draft e.g. FAO Statistic 2014 (line no. 43-44), Leaky et al. 2009 (58-59), Bondeau et al., 2007 (76, 80), Drewniak et al. 2012 (79-80), Lu et al. 2017 (80), Zhao et al. 2007 (168), Meinshausen et al. 2011 (179), Lamarque et al. 2011 (180), Ren et al. 2015 (197), Harris et al., 2014 (212), Viovy, 2016 (212), Meinshausen et al., 2011 (213), Lamarque et al. 2011 (214). 8) Some listed references have missing citation in the draft e.g. Ball & Berry 1987 (line no. 470-472), Chen 1992 (479-480), Drewniak et al. 2013 (488-490), Farquhar et al 1980 (500-501), Gill et al 2014 (505-507), Jonckheere et al. 2004 (508-510), Rajaram et al. 1993 (558-560), Xiaolin R. Weitzel et al. 2013 (595-598). 9) Some word formatting error needs to be corrected e.g. in line no. 110, 111, 382, and 511. 10) In the reference list prescribed journal format should be followed. As the reference of line no. 462-466, 542-545, and 569-571 seems out of the format.

---

## Author Response (AR1)

**"Impact of environmental changes and land-management practices on wheat production in India" by Gahlot et al. submitted to Earth System Dynamics**

**RESPONSE TO COMMENTS BY THE REVIEWERS**

Reviewers' comments are in blue, our responses are in black and the changes made in the document are in green font. The line numbers refer to the marked up document attached herewith.

\_\_\_\_\_

Response to Reviewer 1

\_\_\_\_\_

Based on a dynamic land model, the authors developed the growth processes for spring wheat using field experiments and studied the effects of different environmental factors and land management practices on the spring wheat production in India. The authors have shown that both the increase in CO2 concentration, availability of water through irrigation and additional nitrogen fertilizer enhance the annual wheat production, while the elevated temperature reduces total wheat productions. The authors also investigate the impact of the above factors on wheat production at five spring wheat environment (SWE) regions. The paper is written in a very decent way and the results contribute to our understandings of the impact of induvial environmental factors on wheat production in India.

We thank the reviewer for such encouraging comments.

I have only some minor points to make. First, in the Results section, you discussed changes in the overall wheat production. It might be interesting to also include changes in its components, like gross production and respiration in a table.

This is a very important point. The research work on the development of the spring wheat module was conducted in two parts. This paper deals with the first part where we focus on crop growth features like LAI, biomass and production. Adding GPP and respiration as a table can be done but will not do justice to such a complex issue. We are currently working on the second paper where we study carbon dynamics in spring wheat agroecosystems in a comprehensive manner. That paper will include evaluation of variables like GPP, respiration, NEP etc. against field observations, trends in these variables and how they are affected by environmental and anthropogenic forcings. Adding all these elements will make the current paper unfocussed, unwieldly and long. That is why we prefer not to add carbon fluxes so we can focus on wheat growth and production.

Second, in section 3.3, you analyzed the wheat productions in five SWE regions and their associations with different environmental factors based on the control simulation results. You can also do a similar calculation as done at the country-scale, by comparing results between the control simulation and four sensitivity simulations (that is, you set a constant value to each environmental factor one at a time). In this way, you can probably see the impact of different factors on wheat production at different regions and quantify their overall contributions to the country-scale changes.

We thank the reviewer for this suggestion. This is a valuable suggestion that will add to the understanding of SWEs. We will do the analysis for SWE-specific impact of each external driver on wheat production and add the results in the form of a new table in the paper. Details of this analysis will also be added in the Results and Discussions section. The knowledge added by this additional text will contribute to a comprehensive understanding of wheat production patterns in India.

**Text added in lines 294-305:**

... to identify regions with similar growing conditions for wheat. SWE 1 (Fig. 2) represents mostly the Indo-Gangetic plains that offer good access to irrigation for wheat which is a non-monsoon crop. The growing season temperatures fall in the optimum range for wheat growth. SWE 2, which mainly comprises of the wheat growing regions in the proximity of the Himalayas, is characterized by very low growing season temperatures and high rainfall. SWE 3 represents the north-western parts of the country with moderate to high growing season temperatures, low rainfall and small values of AEI. SWE 4 represents the central parts of India and tropical wheat growing regions with high temperatures and moderate growing season precipitation. SWE 5 represents the crucial wheat growing regions of the country where the

conditions are similar to SWE 1 but irrigation facilities are lacking. Wheat production for each of the SWEs has been discussed further in the following sections.

Text added in lines 462-479:

Similar to the analysis done for country-scale impact of different factors, we quantified the impact of factors on different SWEs. The results of this analysis are summarized in Table 4. SWE1 and SWE5 are the two regions where magnitude of trends in change in wheat production with different input forcings are the highest (Table 4). The magnitudes of impacts of forcings on SWEs 2, 3 and 4 are relatively small. This is because the analysis involves production that is calculated as yield times the harvested area. The numbers in Table 4, hence, do not reflect changes per unit harvested area.

While CO2 fertilization, water added through irrigation and nitrogen fertilizers are found to increase wheat production in SWE1 at 0.26 Mt per ppm [CO2], 0.35Mt per 1000 mm and 0.07 Mt per kg N ha-1 respectively, production is found to decrease by 3.52 Mt for every degree Celsius rise in average growing season temperatures. It is found that water added through irrigation has small yet negative impact on production in SWE2. This can be due to excess surface runoff in SWE2 that might lead to washing away of nitrogen from the soil resulting in nutrient stress in the crop. The impact of different forcings is also found to be significant for SWE5 where [CO2], irrigation and nitrogen fertilizers have promoted wheat production at the rates of 0.07 Mt per ppm [CO2], 0.41 Mt per 1000 mm and 0.01 Mt per kg N ha-1, respectively. Irrigation is seen to have the most impact on wheat production in SWE 5 out of all the SWEs.

**Text added in lines 1109-1114**

| Table 4: Impacts of different e | external forcings | on annual | wheat p | roduction | in the | SWEs |
|---------------------------------|-------------------|-----------|---------|-----------|--------|-------------|
| during the study period (1980-  | -2016).           |           |         |           |        |             |

| Input Forcing (i)             | Change in annual wheat production per unit change in i |                   |                   |                   |                     |
|-------------------------------|---------------------------------------------------------------|-------------------|-------------------|-------------------|---------------------|
|                               | SWE1                                                          | SWE2              | SWE3              | SWE4              | SWE5                |
| Elevated                      | 0.26 a                                             | 0.02 a | 0.01 a | 0.02 a | $0.07^{\mathrm{a}}$ |
| atmospheric CO 2   |                                                               |                   |                   |                   |                     |
| level (MT ppm -1 ) |                                                               |                   |                   |                   |                     |
| Average growing               | -3.52 b                                            | -0.03             | -0.12             | -0.36             | -1.36               |
| season                        |                                                               |                   |                   |                   |                     |
| temperature*                  |                                                               |                   |                   |                   |                     |
| $(Mt \circ C^{-1})$           |                                                               |                   |                   |                   |                     |

| Water demand                                      | 0.35 b | 0.04 b | 0.61 a | 0.07 | 0.41              |
|---------------------------------------------------|-------------------|-------------------|-------------------|------|-------------------|
| $(Mt \ 1000 \ mm^{-1})$                           |                   |                   |                   |      |                   |
| Average nitrogen fertilizer per unit              | 0.07ª             | 0.01              | 0                 | 0    | 0.01 b |
| area
(Mt kg N -1 ha -1 ) |                   |                   |                   |      |                   |

a Values are significant at 99%

b Values are significant at 90%

Response to Reviewer 2

This study uses the land surface model, ISAM, to examine the effect of different environmental factors, including atmospheric CO2, temperature, nitrogen fertilization, and irrigation on spring wheat production in India. First, the authors implemented spring wheat processes in ISAM by updating C3 crop parameterizations. After calibrating and validating the updated model against available observations, ISAM is applied to explore environmental and land management factors on Indian wheat production. It is found that during the last 30 years, increasing atmospheric CO2, addition of nitrogen fertilizer, and irrigation act to increase the production due to increased heat stress. Regional scale analysis of environmental factors and land management practice shows that Indo-Gangetic plain is the best region for growing spring wheat in India, C1 ESDD Interactive comment Printer-friendly version Discussion paper and Northwestern India is the least productive region for wheat growth. This study makes a useful contribution to boost our general understanding in the effect of environmental change and land management on crop yields and production. The manuscript is in general clearly written. I recommend its publication after the following issues are addressed.

We thank the reviewer for the encouraging comments

Lines 61-63: This sentence is hard to read. Please rephrase.

We will modify the following lines in the current paper text:61 Studies that cover the impact of land management practices of irrigation and addition of

62 nitrogen input on crop production aid in giving an overall understanding of the scope of 63 improvement in planting and managing the crop to enhance production (Tack et al. 2017). to:

Quantification of the impact of land management practices on crop production helps in understanding how the croplands can be managed to improve production (Tack et al. 2017). Text changed in lines 73-74:

Quantification of the impacts of land management practices on crop growth helps in understanding how croplands can be managed to improve production.

Line 105: How big is the wheat experimental site?

The experimental plots are approximately 650 sq m. We will add this information to the text. Text added in line 133:

 $\dots$  approximately 650 m2 in area and is  $\dots$

**Lines 139-140: What are these major plant functional types?**

There are 30 PFTs in ISAM. These include different types of forests (e.g., evergreen needleaf deciduous broadleaf, etc.), savannah, croplands, pastures and urban areas. Providing a full list of PFTs is not necessary for this paper because we are focusing only on the croplands area. For brevity, we have provided details that are directly relevant for this study. of the An interested reader can find descriptions of the PFTs and other details in Barman et al. 2014a, 2014b; Song et al. 2013, 2015, 2016; that are cited in the text.

Line 179: Where 'the climate driver' data come from? Also, the reference of Meinshausen et al., 2011 is missing.

The climate data is taken from Climate Research Unit (CRU)-National Centre for Environment Prediction reanalysis (Viovy et al. 2018). The citation for CO2 data will be corrected to Le Quéré et al. 2017 in the text which was previously mentioned as Meinshausen et al., 2011. Line 179 will be changed to:

The model is spun-up by recycling the climate data (CRU data, Viovy et al. 2018),  $[CO_2]$  (Le Quéré et al. 2017) and airborne nitrogen deposition (Lamarque et al. 2011) data for 2015-16

until the soil temperature, soil moisture, the soil carbon pool and the soil nitrogen pool reaches a steady state.

Text added in lines 209-211:

... Climate Research Unit-National Centre for Environment Prediction reanalysis data (CRU-NCEP, Vivoy et al., 2018), Global Carbon Project Budget 2017 [CO2] (Le Quere et al., 2018) ...

Line 225: Section 3.2 Here the effect of a single factor (CO2, temperature, etc.) is obtained by subtracting the simulation that includes the effect of all factors (CTRL) from the simulation that excludes the effect of a certain factor. Thus, the effect includes interactions with other factors. How would it compare with the sole effect of a certain factor by keeping other factors constant? (For example, suppose a simulation in which only atmospheric CO2 changes to represent the CO2 effect). Some discussion on this issue would be helpful.

The current experiment design accounts for how the different input drivers have contributed to the wheat production as we see it today. Since the interactions between the input drivers are nonlinear, and process-based models allow for such interactions, the current experiment design accounts for such non-linearities and interactions. The alternate experiment design, in which only one factor is varied, does not allow such interactions since all other inputs will be kept constant. Such experiment design can be used to study how each input driver can contribute to wheat production, but the results will not be able to match with the actual production numbers in the country because of lack of representation of interactions and feedbacks in the experiment design.

Also, what are the nonlinear interactions among different factors? Does the sum of individual effects add linearly to the combined effect? The authors stated that changes in atmospheric  $CO_2$ , irrigation, fertilizers, and temperature led to 39%, 15%, 20% and -16% changes in countrywide production. So, what explains the residual change in wheat production that are not attributed to these factors? Some discussions should be added.

The Earth System is a nonlinear system where different components interact with each other. We used a process-based model that includes such interactions including interactions between different drivers. For instance, higher temperatures increase the crop water demand. Higher [CO2] increases photosynthesis that also affects nutrient and water demand. Because of these interactions, the sum of the effects will not add up to 100%. Moreover, the experiments are not

exhaustive; there are other factors like relative humidity etc. that affect production. We will add a discussion to this effect in the revised manuscript.

**Text added in lines 561-568:**

The Earth System is a nonlinear system where different components interact with each other. In this study we used a process-based model that includes such interactions including interactions and feedbacks between different drivers. For instance, higher temperatures increase the crop water demand. Higher [CO2] increases photosynthesis that also affects nutrient and water demand. Because of these interactions, the sum of the effects will not add up to 100%. Moreover, the experiments conducted in this study are not exhaustive; there are other factors like relative humidity, solar radiation etc. that might affect production.

**Lines 256-257: '2' -> 'two'**

**The suggested change will be implemented. Text changed in line 325: '2' changed to 'two' Text changed in line 326: '2' changed to 'two'**

**Lines 431-432: This sentence lacks a context. How does this study imply that ISAM will likely provide better estimate of terrestrial carbon flux?**

Currently ISAM uses a generic crop model to simulate all the agroecosystems of India. Using crop-specific models like the spring wheat model developed in this study will improve the simulation of crop phenology over wheat agroecosystems. This will likely lead to better simulations of the spatio-temporal variability in carbon dynamics. We will add this discussion to the Conclusions and Discussions section of the paper.

**Text added in lines 557-559:**

... using crop-specific models like the spring wheat model developed in this study will improve the simulation of crop phenology for agro-ecosystems. This will likely lead to better estimates of carbon fluxes and their spatio-temporal variability.

Table 1: For the experiment  $S_{TEMP}$  that assumes no temperature change, I assume other climate fields such as precipitation and humidity change with time. If so, to what extent changes in other

climate fields such as precipitation and soil moisture contribute to the 'direct' heat stress effect? Some discussions should be provided.

The effects of climate variables of temperature and precipitation have been studied separately in the runs  $S_{TEMP}$  and  $S_{WATER}$  respectively. The  $S_{WATER}$  run, as described in Table 1, allows for no precipitation change and no irrigation. Other climatic variables like humidity are allowed to change with time in all simulations. Soil moisture is calculated in the model based on soil water balance with inputs of precipitation, irrigation, soil type etc.

Table 3: statistic test should be done on the trends shown here.

We will add statistical significance of the results presented in this table. Statistical significance of each trend is computed and the p value is added in Table 3.

Response to Reviewer 3

Comments on the MS The manuscript entitled "Impact of environmental changes and landmanagement practices on wheat production in India" is a very good study to quantify the role of various environmental factors and agricultural management practices on spring wheat production in India during 1980-2016 by Integrated Science Assessment Model (ISAM). Elevated atmospheric CO2 and rising temperature are considered in environmental factors while nitrogen fertilizers applications and water availability through irrigation practices are considered in land management factors. The author's effort is commendable however some minor corrections needed in the draft.

We thank the reviewer for the positive comment.

1) From 1980-2016 for every ppm rise in [CO2] level total wheat production of the country has increased by 0.37 Mt and 39% increase in production compared to the 1980-84 period (Fig. 6a; R2 0.97 while described in the draft R2 0.93) and thus a strong positive correlation has observed. While during the same period total wheat production increased at the rate of 0.10Mt for every kg nitrogen fertilizer-N/ha applied to the farm and 20% increase in production compared to the 1980-84 period (Fig. 6c; R2 0.39), a decrease of 8.38 Mt (\_10% reduction) of wheat per degree Celsius

increase in the average growing season temperature (Fig. 6b; R2 0.26) and 15% increase in production compared to the 1980-84 period due to increased irrigation. (Fig. 6d; R2 0.13). These factors have not shown a strong correlation that needs to improve.

If we understand correctly, the reviewer is asking us to improve the correlation between some of the forcings and the corresponding impact. This cannot be done because the correlations are outcomes of numerical modelling experiments. We cannot improve the values but can provide an explanation. Figure 6 plots the forcings and the impacts at the country-scale. The CO2 forcing is uniform across the country but the others forcings are not. There is strong spatial variability in the temperature, fertilizer and water demand forcings and their impacts. This is why we have developed the SWE approach to explore the forcings at regional scale. These patterns will be clearer when we analyse the data further as suggested by Reviewer 1.

See response to Reviewer 1 point 2

The r2 values mentioned in Fig. 6a is correct. The text will be corrected accordingly. We apologize for this oversight.

Text changed in line 355: '0.93' changed to '0.97'

2) In the draft different equations of the models are not shown e.g. Eq. A1, A2, A3 (line no. 170), Eq. A4, A5 (197); Eq. A6, A7, A8 (205); Eq. A9 (207), Eq. 10 (233), Eq. 11 (259) or may be included in the additional/supplement materials.

All the equations are already included in the supplementary material.

3) Data for the actual amount of water used for irrigation is not available. So in the SCON simulation, every grid cell is considered 100% irrigated and crops do not undergo water stress at any point in the growing season and all the regions are 100% irrigated. Since wheat is a non-monsoon crop, is highly dependent on the availability of irrigation. The development of irrigation water use datasets could reduce the uncertainty in simulating the effect of water stress on crop production.

We agree with the reviewer's comment that irrigation water use datasets can help in reducing the uncertainty in simulating the effect of water stress on crop production. We have already mentioned

this in the Conclusions and Discussions sections (lines 438-444). Developing such a dataset is a huge task that is beyond the scope of this study.

4) Variation in wheat productivity in different regions as well as in different years of the study period (1980-2016) depends not only on environmental factors and management practices but also on the genetic factors, multiple cropping's, insect pests and diseases. Since 1980 various hybrids and high yielding wheat varieties were cultivated to increase the input use efficiency and higher economic yield. Similarly, in different climatic zones, area-specific resistant wheat varieties were also grown to enhance wheat productivity. The addition of new processes accounting for the effects of pests, multiple cropping and genotypes will make the simulations more representative of the Indian situation.

We agree with the reviewer's comment that accounting for additional effects like pests, multiple cropping and modelling more genotypes will ensure better representation of the actual scenario in the simulations. We have already discussed this in the Conclusions and Discussions sections (lines 435-438). Because the model developed and used in this study is a process-based model, implementation of every process requires data. We do not have adequate data in the public domain from Indian wheat ecosystems that can be used for this purpose.

5) The study is more generalized for different climatic zones/spring wheat environment (SWE) while there is a need for more focused regional-scale studies. However, the study is an attempt to work in the similar direction with a focus on wheat in India.

We appreciate the reviewer's comment. We decided to introduce the concept of SWEs to address the spatial variability in the forcings and impacts.

6) In the draft multiple citations should be arranged in descending order of the publication year (line no. 54, 58-59, 68-69, 76, 79-80, 83, 137, 145, 212, 270-271, 277-278, 303) and citation of line no. 76, 212, 213 needs to correct as per the formatting guidelines of the journal.

We apologize for these oversights. We will make corrections in the manuscript to ensure proper citation of the references.

All citations are now in chronologically descending order.

7) References missing for some of the citations in the draft e.g. FAO Statistic 2014 (line no. 43-44), Leaky et al. 2009 (58-59), Bondeau et al., 2007 (76, 80), Drewniak et al. 2012 (79-80), Lu et al. 2017 (80), Zhao et al. 2007 (168), Meinshausen et al. 2011 (179), Lamarque et al. 2011 (180), Ren et al. 2015 (197), Harris et al., 2014 (212), Viovy, 2016 (212), Meinshausen et al., 2011 (213), Lamarque et al. 2011 (214).

We apologize for these oversights. We will re-check and correct the reference list and the citations in the text.

The citations have been matched with the list of references. References are formatted as per ESD guidelines with journal acronyms from the Caltech library..

8) Some listed references have missing citation in the draft e.g. Ball & Berry 1987 (line no. 470-472), Chen 1992 (479-480), Drewniak et al. 2013 (488-490), Farquhar et al 1980 (500-501), Gill et al 2014 (505-507), Jonckheere et al. 2004 (508-510), Rajaram et al. 1993 (558-560), Xiaolin R. Weitzel et al. 2013 (595-598). 9) Some word formatting error needs to be corrected e.g. in line no. 110, 111, 382, and 511. 10) In the reference list prescribed journal format should be followed. As the reference of line no. 462-466, 542-545, and 569-571 seems out of the format.

We apologize for these oversights. We will re-check and correct the reference list and the citations in the text.

The citations have been matched with the list of references. References are formatted as per ESD guidelines using journal acronyms from the Caltech library.

We draw the attention of the editor to the following two issues that arose while responding to the comments of Reviewer 2.

 Reviewer 2 had asked for statistical significance in the trends of the impacts of the 4 drivers. In the earlier version, we had calculated linear trends. But in the revised version, we used Sen's slope that is a more robust measure and its statistical significance. The new method leads to small but non-trivial changes in the trend values. These changed values are reported in lines 27-33 (Abstract) and Table 3 in the revised manuscript.

Line 27: '0.68' changed to '0.67', '0.24' changed to '0.25' and '0.31 changed to '0.26' Line 32: '0.37' changed to '0.39' Table 3 Row 2 Column 3: '0.68' changed to '0.67' Table 3 Row 3 Column 3: '0.37' changed to '0.39' Table 3 Row 4 Column 2: '442.50' changed to '443.94' Table 3 Row 4 Column 3: '0.24' changed to '0.25' Table 3 Row 4 Column 4: '0.44' changed to '0.31' Table 3 Row 5 Column 2: '2.66' changed to '2.71' Table 3 Row 5 Column 3: '0.31' changed to '0.26'

 Reviewer 2 had asked for a discussion about the percentage impacts. In the earlier version of the manuscript, we had calculated percentage effects of the different forcings as requested by the editor using the following formula:

**% effect of factor X =**

Change in production due to all factors – change in production due to factor X Production due to **factor X**

× 100%

In this revised version, we use the following formula:

% effect of factor X =

Change in production due to all factors – change in production due to factor X Production due to **all factors**

× 100%

This formulation is more appropriate because it allows for a direct comparison between the effects of different factors and help us to clarify the issue raise by Reviewer 2. However, it leads to changes in the percentage values reported earlier. The changed values are reported in lines 34 (Abstract), 339, 365, 375, 397 (Results), 495, 504, 510 and 521 (Conclusions) in the revised manuscript.

Line 34:'39%' changed to '30%', '15%' changed to '12%', '20%' changed to '15%' and '-16%' changed to '-18%'

Line 341: '39%' changed to '30%'

Line 367: '20%' changed to '15%'

- Line 377: '15%' changed to '12%'
- Line 400: '16%' changed to '18%'
- Line 498: '39%' changed to '30%'
- Line 507: '20%' changed to '15%'
- Line 513: '15%' changed to '12%'
- Line 524: '16%' changed to '18%'

**Impact of environmental changes and land-management practices on wheat production in India**

Shilpa Gahlot1, Tzu-Shun Lin2, Atul K Jain2, Somnath Baidya Roy1, Vinay K Sehgal3,
 Rajkumar Dhakar3

1Centre for Atmospheric Science, Indian Institute of Technology Delhi, New Delhi, 110016
 7 India,

8 2Department of Atmospheric Science, University of Illinois, Urbana, IL, 61801 USA,

3Department of Agricultural Physics, Indian Agricultural Research Institute, New Delhi,
 110012, India

11 *Correspondence to*: Somnath Baidya Roy (drsbr@iitd.ac.in)

1

1

2

12 Abstract. Spring wheat is a major food crop that is a staple for a large number of people in 13 India and the world. To address the issue of food security, it is essential to understand how 14 productivity of spring wheat changes with changes in environmental conditions and agricultural management practices. The goal of this study is to quantify the role of different 15 16 environmental factors and management practices on wheat production in India in recent years (1980 to 2016). Elevated atmospheric CO2 concentration ([CO2]) and climate change are 17 18 identified as two major factors that represent changes in the environment. The addition of 19 nitrogen fertilizers and irrigation practices are the two land-management factors considered in 20 this study. To study the effects of these factors on wheat growth and production, we developed 21 crop growth processes for spring wheat in India and implemented them in the Integrated Science Assessment Model (ISAM), a state-of-the-art land model. The model is able to capture 22 23 site-level observed crop leaf area index (LAI) and country scale production. Numerical experiments are conducted with the model to quantify the effect of each factor on wheat 24 25 production on a country scale for India. Our results show that elevated [CO2] levels, water availability through irrigation and nitrogen fertilizers have led to an increase in annual wheat 26 production at 0.67, 0.25 and 0.26 Mt yr-1, respectively, averaged over the time period 1980-27

| Deleted: 68            |  |
|------------------------|--|
| Deleted: 24            |  |
| Deleted: 31            |  |
| Deleted: /             |  |
| Formatted: Superscript |  |

32 2016. However, elevated temperatures have reduced the total wheat production at a rate of 0.3933 Mt  $yr^{-1}$  during the study period. Overall, the [CO2], irrigation, fertilizers, and temperature 34 forcings have led to 22Mt (30%), 8.47 Mt (12%), 10.63 Mt (15%) and -13 Mt (-18%) changes in countrywide production, respectively. The magnitudes of these factors spatially vary across 35 the country thereby affecting production at regional scales. Results show that favourable 36 37 growing season temperatures, moderate to high fertilizer application, high availability of irrigation facilities, and moderate water demand make the Indo-Gangetic plain the most 38 productive region while the arid northwest region is the least productive due to high 39 40 temperatures and lack of irrigation facilities to meet the high water demand.

**Deleted: 37**

| Deleted: /             |  |
|------------------------|--|
| Formatted: Superscript |  |
| Deleted: 9             |  |
| Deleted: 5             |  |
| Deleted: 20            |  |
| Deleted: 6             |  |

**42 **1 Introduction**

41

43 Wheat is a major food crop, ranked third in India and fourth in the world in terms of its production (FAOSTAT, 2019). Wheat can be of two main types: winter and spring wheat. 44 45 Winter wheat undergoes a 30-40 day long vernalization period induced by below-freezing temperatures and hence has a longer growing season of 180-250 days. In contrast, spring wheat, 46 47 which does not undergo vernalization, has a growing season of 100-130 days (FAO Crop 48 Information, 2018). In India, spring wheat is sown during October-November and harvested 49 during February-April (Sacks et al., 2010). It is grown in widely divergent climatic conditions across the country where different environmental factors like temperature, water availability, 50 51 and [CO2] may affect growth and yield. Ideally, a daily average temperature range of 20-25 °C is ideal for wheat growth (MOA, 2016). Studies have reported heat stress in wheat for 52 53 temperatures between 25 °C to 35 °C (Deryng et al., 2014) during the grain development stages. 54 Beyond the temperatures of 35 °C, wheat fails to survive. High temperatures are terminal to 55 wheat yield specifically in the flowering and grain filling stages during the second half of the 56 growing season (Farooq et al., 2011). Increasing temperature change and heat stress events in

65 the recent decades and their impacts on wheat crop growth processes are extensively studied (Asseng et al., 2015; Lobell et al., 2012; Farooq et al., 2011; Ortiz et al. 2008). Another 66 67 environmental factor that has been widely studied is the impact of increasing [CO2]. The resulting CO2 fertilization effect is found to promote crop growth (Dubey et al., 2015). Apart 68 from environmental factors, management practices including nitrogen fertilizer application and 69 70 irrigation also significantly affect wheat production (Myers et al., 2017; Leaky et al., 2009; 71 Luo et al., 2009). Because wheat is grown in the non-monsoon months, it is a high irrigation 72 crop with almost 94% of the wheat fields in India equipped for irrigation (MAFW, 2017). 73 Ouantification of the impacts of land management practices on crop growth helps in 74 understanding how croplands can be managed to improve production (Tack et al. 2017).

75 Even though India is the third largest wheat producer in the world, domestic production is 76 barely sufficient to meet the country's demand for food and livestock feed (USDA, 2018). Data 77 from different sources report a relatively poor yield of wheat in India as compared to other 78 countries (FAOSTAT, 2019). Hence, there is an urgent need to address this yield gap by developing better land-management practices under different environmental conditions 79 80 (Stratonovitch et al. 2015; Zhao et al. 2014; Luo et al., 2009). A key first step to achieve this goal is to understand the processes involved in interactions of the crop with its environment 81 82 and the factors responsible for impacting crop growth.

Dynamic Growth Vegetation Models (DGVMs) are well-established tools to study global climate-vegetation systems. Implementation of crop-specific parameterization and processes in DGVMs provides us with a better framework to assess and represent the role of agriculture in climate-vegetation systems (Song et al., 2013; Bondeau et al., 2007). This helps in better estimation of biogeochemical and biogeophysical processes, improves the representation of feedback mechanisms as well as prediction of yield and production. Multiple process-based

[revised manuscript text omitted]

**256 257**

255

**258 **2.6 Country-scale simulations**

decadal data from Siebert et al. (2015) (Eq. A9).

Country-scale simulations are conducted after model calibration and evaluation. First, we spin 259 260 up the model for the year 1901 by repeating the climate forcing data of CRU-NCEP (Viovy, (2018) for the period 1901-1920, and fixed year (1901) data for  $[CO_2]$  of 296.8 ppm and data 261 for airborne nitrogen deposition (Dentener, 2006), and zero amount of nitrogen fertilizer and 262 irrigation, until the soil temperature, soil moisture and the soil carbon and nitrogen pools reach 263 a steady state at approximately 1901 levels. Details of the spin-up process are described in 264 265 detail in Gahlot et al. (2017). After the model spin-up, numerical experiments are conducted as transient runs from 1901 to 2016. To estimate the effects of external forcings, country-scale 266 267 runs are conducted over wheat-growing regions in India by varying different input forcings 268 (Table 1). Control run ( $S_{CON}$ ) represents the model run from 1901 to 2016 with time-varying annual [CO2], climate data, annual grid-specific nitrogen fertilizer, and full irrigation to fulfil 269 270 the water needs of the crop. Four additional simulations are conducted by assigning a constant 271 value to each input forcing one at a time. For instance, in SCO2, all input variables (temperature, 272 nitrogen, and irrigation) are the same as in the SCON case except [CO2] that is held constant at 273 1901 level. The difference in model simulations from SCON and SCO2 then gives the effect of 274 elevated [CO2] on wheat crop growth processes. Here we present the results only for the recent 275 decades. 1980 to 2016. 276

A8). Annual Area Equipped for Irrigation (AEI) dataset is created by Jinear interpolation of

[revised manuscript text omitted]

CO2 fertilization is the most dominant factor that has contributed to increase in wheat production over India. Annual average [CO2] worldwide has increased from 337.7 ppm in 1980 to 404.3 ppm in 2016. This increase in levels of [CO2] at the rate of 1.82 ppm yr-1 has promoted growth in wheat as elevated [CO2] levels are known to enhance photosynthetic CO2 fixation and have a positive impact on most C3 plants (Myers et al. 2017; Leakey et al. 2009; Allen et al., 1996). Our results show that for every ppm rise in [CO2] level total wheat production of the country has increased by 0.37 Mt (Fig. 6a; Table 3). This amounts to a 22 Mt (30%) increase

**11**

[revised manuscript text omitted]

| Deleted: Ball, J. T., Woodrow, I. E., & Berry, J. A.: A
model predicting stomatal conductance and its contribution |  |  |
|-----------------------------------------------------------------------------------------------------------------------|--|--|
| Deleted: &                                                                                                            |  |  |
| Deleted: al                                                                                                           |  |  |
| Deleted: ogy                                                                                                          |  |  |
| Deleted: (5)                                                                                                          |  |  |
| Deleted: 2014a. https://doi.org/10.1111/gcb.12474                                                                     |  |  |
| Deleted: &                                                                                                            |  |  |
| Deleted: al                                                                                                           |  |  |
| Deleted: ogy                                                                                                          |  |  |
| Deleted: (6)                                                                                                          |  |  |
| Deleted: 2014b.                                                                                                       |  |  |
| Deleted: Chen, J. M., & Black, T. A.: Defining leaf areq2                                                             |  |  |
| Deleted: &                                                                                                            |  |  |
| Deleted: Int. J. Curr. Microbiol. App. Sci.,                                                                          |  |  |
| Deleted: (1)                                                                                                          |  |  |
| Deleted: ¶                                                                                                            |  |  |
| Formatted: English (UK)                                                                                               |  |  |
| Deleted: &                                                                                                            |  |  |
| Deleted: ironmental                                                                                                   |  |  |
| Deleted: earch                                                                                                        |  |  |
| Deleted: ers                                                                                                          |  |  |
| Deleted: (3)                                                                                                          |  |  |
| Deleted: Directorate of Economics and Statistics, [3]                                                                 |  |  |
| Deleted: . https://doi.org/10.5194/gmd-6-495-2013                                                                     |  |  |
| Deleted: &                                                                                                            |  |  |
| Deleted: ical                                                                                                         |  |  |
| Deleted: iews                                                                                                         |  |  |
| Deleted: in                                                                                                           |  |  |
| Deleted: ironmental                                                                                                   |  |  |
| Deleted: ence and                                                                                                     |  |  |
| Deleted: nology                                                                                                       |  |  |
|                                                                                                                       |  |  |
| Deleted: (21)                                                                                                         |  |  |

| 692 | FAO Crop Information . http://www.fao.org/land-water/databases-and-software/crop-         | < (         | Dele                |
|-----|--------------------------------------------------------------------------------------------------|-------------|---------------------|
| 693 | information/wheat/en/ Jast access: 15, November 2018,                                     | (           | Dele                |
| 694 | Farooq, M., Bramley, H., Palta, J. A., and Siddique, K. H.: Heat stress in wheat during          | $\square$   | Dele                |
| 695 | reproductive and grain-filling phases, Crit, Rev. Plant Sci., 30, 491-507, 2011.                 | $\setminus$ | Dele                |
| 696 | Gablot S Shu S Jain A K and Baidya Roy S: Estimating trends and variation of net                 | NY          | Dele                |
| (07 | biene meductivity in India for 1000, 2012 yeing a land surface model. Coording Bas Latt          | ()  Y       | Dele                |
| 697 | biome productivity in India for 1980–2012 using a fand surface model, Geophys, Res, Lett.        | )////(      | Dele                |
| 698 | 44, https://doi.org/10.1002/2017GL075777, 2017.                                           |             | Dele                |
| 699 | Kimball, B. A. Crop responses to elevated CO2 and interactions with H2O, N, and                  |             | Dele                |
| 700 | temperature. Curr * Opin * Plant Biol * 31, 36–43, 2016. |             | Dele
J. A.:      |
| 701 | Koehler, A. K., Challinor, A. J., Hawkins, E., and Asseng, S.; Influences of increasing          |             | assin
1980       |
| 702 | temperature on Indian wheat: quantifying limits to predictability, Environ. Res. Lett., 8,       |             | Dele                |
| 703 | 034016, 2013.                                                                                    |             | Dele                |
| 704 | Le Quéré, C., Andrew, R. M., Friedlingstein, P., Sitch, S., Pongratz, J., Manning, A. C.,        |             | Dele                |
| 705 | Korsbakken, J. I., Peters, G. P., Canadell, J. G., Jackson, R. B., Boden, T, Tans, P. P.,        |             | Dele                |
| 706 | Andrews, O. D., Arora, V. K., Bakker, D. C. E., Barbero, L., Becker, M., Betts, R. A.,           |             | Field               |
| 707 | Bopp, L., Chevallier, F., Chini, L. P., Ciais, P., Cosca, C. E., Cross, J., Currie, K., Gasser,  |             | Dele
Sand |
| 708 | T., Harris, I., Hauck, J., Haverd, V., Houghton, R. A., Hunt, C. W., Hurtt, G., Ilyina, T.,      |             | Dele                |
| 709 | Jain, A. K., Kato, E., Kautz, M., Keeling, R. F., Klein Goldewijk, K., Körtzinger, A.,           |             | Dele                |
| 710 | Landschützer, P., Lefèvre, N., Lenton, A., Lienert, S., Lima, I., Lombardozzi, D., Metzl,        |             | Dele                |
| 711 | N., Millero, F., Monteiro, P. M. S., Munro, D. R., Nabel, J. E. M. S., Nakaoka, SI., Nojiri,     |             | Dele                |
| 712 | Y., Padin, X. A., Peregon, A., Pfeil, B., Pierrot, D., Poulter, B., Rehder, G., Reimer, J.,      |             | Dele                |
| 713 | Rödenbeck, C., Schwinger, J., Séférian, R., Skjelvan, I., Stocker, B. D., Tian, H., Tilbrook,    |             | Dele                |
| 714 | B., Tubiello, F. N., van der Laan-Luijkx, I. T., van der Werf, G. R., van Heuven, S., Viovy,     |             | Dele                |
| 715 | N., Vuichard, N., Walker, A. P., Watson, A. J., Wiltshire, A. J., Zaehle, S., and Zhu, D.:       | Ć           | Dele                |
| 716 | Clabel Cash on Dudget 2017 Forth State Sci Date 10, 405, 449, 2019                               | (           | Dele                |
| /16 | Giodai Cardon Budget 2017, Earth Syst. Sci. Data, 10, 405–448, 2018.                      | (           | Dele                |

| Deleted:                                                                                                                                                                                        |
|-------------------------------------------------------------------------------------------------------------------------------------------------------------------------------------------------|
| Deleted: statistic                                                                                                                                                                              |
| Deleted: accessed                                                                                                                                                                               |
| Deleted: th                                                                                                                                                                          |
| Deleted: , 2014                                                                                                                                                                          |
| Deleted: &                                                                                                                                                                                      |
| Deleted: ical                                                                                                                                                                                   |
| Deleted: iews in                                                                                                                                                                                |
| Deleted: ences                                                                                                                                                                                  |
| Deleted: (6)                                                                                                                                                                                    |
| Deleted: Farquhar, G. V., von Caemmerer, S. V., & Berry, J. A.: A biochemical model of photosynthetic CO 2 assimilation in leaves of C3 species. Planta, 149(1), 78-90, 1980. |
| Deleted: &                                                                                                                                                                                      |
| Deleted: ical                                                                                                                                                                                   |
| Deleted: earch                                                                                                                                                                                  |
| Deleted: ers                                                                                                                                                                                    |
| Deleted: 2017.                                                                                                                                                                                  |
| Field Code Changed                                                                                                                                                                              |
| Deleted: Gill, K. K., Babuta, R., Kaur, N., Kaur, P., &
Sandhu, S. S.: Thermal requirement of wheat crop in [4]                                                                              |
| Deleted: (2016)                                                                                                                                                                                 |
| Deleted: .                                                                                                                                                                                      |
| Deleted: ent                                                                                                                                                                                    |
| Deleted: ion                                                                                                                                                                                    |
| Deleted: in                                                                                                                                                                                     |
| Deleted: ogy                                                                                                                                                                                    |
| Deleted: . doi: 10.1016/j.pbi.2016.03.006                                                                                                                                                       |
| Deleted: &                                                                                                                                                                                      |
| Deleted: ,                                                                                                                                                                                      |
| Deleted: Environmental                                                                                                                                                                          |
| Deleted: Research                                                                                                                                                                               |
| Deleted: ers                                                                                                                                                                                    |
| Deleted: (3)                                                                                                                                                                                    |

| 760  | Leakey, A. D., Ainsworth, E. A., Bernacchi, C. J., Rogers, A., Long, S. P., and Ort, D. R.:      |              | Del  |
|------|--------------------------------------------------------------------------------------------------|--------------|------|
| 761  | Elevated CO2 effects on plant carbon, nitrogen, and water relations: six important lessons       |              | Glo  |
| 762  | from FACE, J. Exp. Bot., 60, 2859-2876, 2009.                                                    |              | Disc |
| 763  | Lobell, D. B., Sibley, A., and Ortiz-Monasterio, J. I.: Extreme heat effects on wheat senescence |              | Del  |
| 764  | in India, Nat, Clim, Change, 2, 186- 189 , 2012.                                          | \            | Del  |
| 765  | Lokupitiya, E., Denning, S., Paustian, K., Baker, I., Schaefer, K., Verma, S., Meyers, T.,       | MM           | Del  |
| 766  | Bernacchi C. J. Suyker, A. and Fischer, M. Incorporation of crop phenology in Simple             |              | Del  |
| 1/00 | Directory Model (CiDerer) to improve load strengther and a series from                           |              |      |
| /6/  | Biosphere Model (SiBcrop) to improve land-atmosphere carbon exchanges from                       |              | Del  |
| 768  | croplands, Biogeosci ences , 6, 969-986, 2009.                                            |              | Del  |
| 769  | Lu, Y., Williams, I. N., Bagley, J. E., Torn, M. S., and Kueppers, L. M.: Representing winter    |              | Del  |
| 770  | wheat in the Community Land Model (version 4.5). Geosci. Model Dev., 10, 1873-1888,              | - \\ {       | Del  |
| 771  | 2017.                                                                                            |              | Del  |
| 772  | Luo, Q., Bellotti, W., Williams, M., and Wang, E.: Adaptation to climate change of wheat         |              | Del  |
| 773  | growing in South Australia: analysis of management and breeding strategies, Agr. Ecosyst.        |              | Del  |
| 774  | Environ, 129, 261-267, 2009.                                                                     | (            | Del  |
| 775  | MAFW: Agricultural Statistics at a Glance 2016. Directorate of Economics and Statistics.         | $\mathbb{N}$ | Del  |
| 776  | Ministry of Agriculture Government of India PDES-256 (F) 500-2017 - (DSK-III)                    |              | Del  |
| 170  | https://conde.deenet.nie.in/DDE/Clance.2016.ndf.2017                                             |              | Del  |
| ///  | nups.//eands.daenet.mc.m/PDP/Ofance-2010.pdf, 2017.                                              |              | Del  |
| 778  | Maiorano, A., Martre, P., Asseng, S., Ewert, F., Müller, C., Rötter, R. P., Ruane, A. C.,        |              | Del  |
| 779  | Semenov, M. A., Wallach, D., Wang, E., Alderman, P. D., Kassie, B. T., Biernath, C.,             | N Y          | Del  |
| 780  | Basso, B., Cammarano, D., Challinor, A. J., Doltra, J., Dumont, B., Rezaei, E. E., Gayler,       |              | Dire |
| 781  | S., Kersebaum, K. C., Kimball, B. A., Koehler, A. K., Liu, B., O'Leary, G. J., Olesen, J.        | (
(       | Mov  |
| 782  | E., Ottman, M. J., Priesack, E., Reynolds, M., Stratonovich, P., Streck, T., Thorburn, P. J.,    | (            |      |
| 783  | Waha, K., Wall, G. W., White, J. W., Zhao, Z., Zhu, Y.: Crop model improvement reduces           |              | Del  |
| 784  | the uncertainty of the response to temperature of multi-model ensembles. Field Cron              |              | Del  |
| 785  | Pec 202 5.20 2017                                                                                | (            | Del  |
| 105  | 103, 202, J-20, 2011.                                                                     | £            | Del  |

|                                   | Deleted: &                                                                                                                  |
|-----------------------------------|-----------------------------------------------------------------------------------------------------------------------------|
| N                                 | Deleted: ournal of                                                                                                          |
| W                                 | Deleted: experimental                                                                                                       |
| ())                               | Deleted: b                                                                                                                  |
| M                                 | Deleted: any                                                                                                                |
|                                   | Deleted: (10)                                                                                                        |
|                                   | Deleted: &                                                                                                                  |
| $( \parallel)$                    | Deleted: ure                                                                                                                |
| $\langle \rangle \rangle$         | Deleted: ate                                                                                                                |
| $\langle \rangle \rangle$         | Deleted: (3)                                                                                                                |
| $\langle \rangle \langle \rangle$ | Deleted: &                                                                                                                  |
|                                   | Deleted: ences                                                                                                              |
| 1                                 | Deleted: (6)                                                                                                                |
|                                   | Formatted: English (UK)                                                                                                     |
|                                   | Deleted: &                                                                                                                  |
|                                   | Deleted: i                                                                                                                  |
| (l)                               | Deleted: culture,                                                                                                           |
| M/                                | Deleted: e                                                                                                                  |
| $\parallel /$                     | Deleted: tems &                                                                                                             |
| $\langle \rangle \rangle$         | Deleted: e                                                                                                                  |
| $\langle \rangle$                 | Deleted: ironment                                                                                                           |
| $\langle \rangle$                 | Deleted: (1-3)                                                                                                       |
| $\langle \rangle$                 | Deleted: MAFW: District-wise Crop Production Statistics,
Directorate of Economics and Statistics, Ministry of [5] |
|                                   | Deleted: 2017. Retrieved from                                                                                               |
| /                                 | Moved (insertion) [1]                                                                                                       |
|                                   | Deleted: & Alderman P. D.                                                                                            |
|                                   | Deleted: c                                                                                                                  |
|                                   | Deleted: s                                                                                                                  |
|                                   | Deleted: r                                                                                                                  |
|                                   | Deleted: earch                                                                                                              |
|                                   | \                                                                                                                           |

| 824 | MOA: Status Paper on Wheat, Directorate of Wheat Development, Ministry of Agriculture,         |                                                                                                                    |
|-----|------------------------------------------------------------------------------------------------|--------------------------------------------------------------------------------------------------------------------|
| 825 | Govt. of India, 180 pp, https://www.nfsm.gov.in/StatusPaper/Wheat2016.pdf, 2016.               | Deleted: Department of Agriculture and Co-operation,
U.P.,ovt. of India, 180 pp, 2016.                          |
| 826 | Monfreda, C., N. Ramankutty, and J. A. Foley: Farming the planet: 2. Geographic distribution   | Moved up [1]: Maiorano, A., Martre, P., Asseng, S.,                                                                |
| 827 | of crop areas, yields, physiological types, and net primary production in the year 2000,       | Ewert, F., Müller, C., Rötter, R. P., & Alderman, P. D.:
Crop model improvement reduces the uncertainty of the  |
| 828 | Global Biogeochem, Cy, 22, GB1022, https://doi.org/10.1029/2007GB002947, 2008.                 | response to temperature of multi-model ensembles, Field crops research, 202, 5-20, 2017.                           |
| 829 | Mueller, N. D., Gerber, J. S., Johnston, M., Ray, D. K., Ramankutty, N., and Foley, J. A.:     | Deleted: icalCy.cles 22, GB1022, 2008                                                                       |
| 830 | Closing yield gaps through nutrient and water management, Nature, 490, 254-257, 2012.          | https://                                                                                                           |
| 831 | Myers, S. S., Smith, M. R., Guth, S., Golden, C. D., Vaitla, B., Mueller, N. D., Dangour, A.   | Deleted: & nd Foley, J. A.: Closing yield gaps through nutrient and water management, Nature, 490(7419) [8] |
| 832 | D., and Huybers, P.: Climate change and global food systems: potential impacts on food         | Deleted: &angour, A. D., and Huybers, P.: Climate                                                                  |
| 833 | security and undernutrition. Annu, Rev, Publ, Health, 38, 259-277, 2017.                       | change and global food systems: potential impacts on food security and undernutrition. Annu.ualRrv.iewPof          |
| 834 | NFSM, Crop Calendar by National Food Security Mission (NFSM), Ministry of Agriculture          | pbl.icHh [9]                                                                                                       |
| 835 | and Farmers Welfare, Government of India, https://nfsm.gov.in/nfmis/rpt/                       | Deleted:https://nfsm.gov.in/nfmis/rpt/                                                                             |
| 836 | calenderreport.aspx, last access; 5th January 2018.                                            |                                                                                                                    |
| 837 | Ortiz, R., Sayre, K. D., Govaerts, B., Gupta, R., Subbarao, G. V., Ban, T., Hodson, D., Dixon, |                                                                                                                    |
| 838 | J. M., Ortiz-Monasterio, J. I., and Reynolds, M.: Climate change: Can wheat beat the           | Deleted: ∧ Reynolds, M.: Climate change: Ccn                                                                       |
| 839 | heat?, Agr, Ecosyst, Environ, 126, 46-58, 2008.                                                | wheat beat the heat?, Agriculture,Ecosyst.tems         &Environ.ironment 126(1-2)         [11]                     |
| 840 | Rajaram, S., Van Ginkel, M., and Fischer, R. A.: CIMMYT's wheat breeding mega-                 | Deleted: &                                                                                                         |
| 841 | environments (ME), Proceedings of the 8th International wheat genetic symposium, 1101-         |                                                                                                                    |
| 842 | 1106, 1993.                                                                                    |                                                                                                                    |
| 843 | Rosenzweig, C., Elliott, J., Deryng, D., Ruane, A. C., Müller, C., Arneth, A., Boote, K. J.,   |                                                                                                                    |
| 844 | Folberth, C., Glotter, M., Khabarov, N., Neumann, K., Piontek, F., Pugh, T. A. M., Schmid,     |                                                                                                                    |
| 845 | E., Stehfest, E., Yang, H., and Jones, J. W. Assessing agricultural risks of climate change    | Deleted: & Neumann, K Assessing agricultural                                                                |
| 846 | in the 21st century in a global gridded crop model intercomparison, P. Natl, Acad, Sci.        | gridded crop model intercomparison, Proceedings of the                                                             |
| 847 | USA, 111, 3268-3273, 2014.                                                                     | Nati.ionaiAcademy of Sci. USAences 111(9) [12]                                                                     |
| 848 | Sen, P. K.: Estimates of the regression coefficient based on Kendall's tau, J. Am, Stat,       | Deleted: Journal of the                                                                                            |
| 849 | Assoc. 63, 1379-1389, 1968.                                                                    | Am.ericanSsat.isticalAasoc.ociation 63(3243)                                                                       |
| I   |                                                                                                |                                                                                                                    |

[revised manuscript text omitted]